# The formation of unsaturated $IrO_x$ in $SrIrO_3$ by cobalt-doping for acidic oxygen evolution reaction

Jia-Wei Zhao[1,2,3,7], Kaihang Yue [4,7], Hong Zhang[5,7], Shu-Yin Wei[2], Jiawei Zhu[3], Dongdong Wang[3], Junze Chen[1], Vyacheslav Yu. Fominski[6] & Gao-Ren Li [1] ✉

Electrocatalytic water splitting is a promising route for sustainable hydrogen production. However, the high overpotential of the anodic oxygen evolution reaction poses significant challenge. $SrIrO_3$-based perovskite-type catalysts have shown great potential for acidic oxygen evolution reaction, but the origins of their high activity are still unclear. Herein, we develop a Co-doped $SrIrO_3$ system to enhance oxygen evolution reaction activity and elucidate the origin of catalytic activity. In situ experiments reveal Co activates surface lattice oxygen, rapidly exposing $IrO_x$ active sites, while bulk Co doping optimizes the adsorbate binding energy of $IrO_x$. The Co-doped $SrIrO_3$ demonstrates high oxygen evolution reaction electrocatalytic activity, markedly surpassing the commercial $IrO_2$ catalysts in both conventional electrolyzer and proton exchange membrane water electrolyzer.

Water splitting for hydrogen production offers the advantage of producing clean and sustainable fuel without carbon emissions[1,2]. To date, proton exchange membrane (PEM) water electrolysis is one of the most established ways in the field of green hydrogen production[3–5]. However, the high overpotential typically associated with the anodic oxygen evolution reaction (OER) poses a significant challenge to enhancing hydrogen production efficiency from water electrolysis[6]. Furthermore, the advancement of OER catalysts designed for acidic medium poses a greater challenge compared to those intended for alkaline medium. Since highly OER-active electrocatalysts are mainly comprised of metal oxides or hydroxides, most of which exhibit poor stability under acidic conditions[7–10]. Therefore, the development of efficient OER catalysts functioning in acidic medium is critical.

Current, effective acidic OER catalysts include Ru, Ir, and Mn-based metal oxides[11–17]. In particular, $SrIrO_3$ with an $ABO_3$ perovskite structure (where A typically represents an alkaline earth metal and B represents a transition metal) shows high acidic OER catalytic performance[18–30]. In recent years, extensive researchers have been conducted to understand the OER catalytic mechanism of $SrIrO_3$. Jaramillo et al. prepared the (001) plane $SrIrO_3$ film via a laser epitaxy strategy, and observed an incremental improvement in its OER catalytic performance throughout the catalytic process[22]. Through combining theoretical calculations and experiments, they discovered that the high catalytic activity of $SrIrO_3$ could be attributed to the exposed $IrO_x$ sites following Sr dissolution. In further comprehensive studies, researchers confirmed the structural modifications of $SrIrO_3$ under acidic OER process using secondary ion mass spectrometry (SIMS), in situ atomic force microscopy (AFM), and X-ray absorption spectroscopy (XAS), respectively[23–25]. They proposed a correlation between the Sr dissolution process and the formed $IrO_x$ surface activities. Moreover, an investigation involving $SrIr_{0.1}Co_{0.9}O_3$ further indicates that the OER activity also originates from the amorphous $IrO_x$ structure formed by the dissolution of Co[26]. These studies appear to identify the active component of $SrIrO_3$-based perovskites as the amorphous

[1]College of Materials Science and Engineering, Sichuan University, Chengdu 610065, China. [2]School of Chemistry, Sun Yat-sen University, Guangzhou 510275, China. [3]Department of Mechanical Engineering, City University of Hong Kong, 83 Tat Chee Avenue, Kowloon, Hong Kong SAR 999077, China. [4]CAS Key Laboratory of Materials for Energy Conversion, Shanghai Institute of Ceramics, Chinese Academy of Sciences (SICCAS), 585 Heshuo Road, Shanghai 200050, China. [5]Electron Microscopy Centre, School of Physical Science and Technology, Lanzhou University, Lanzhou 730099, China. [6]National Research Nuclear University MEPhI (Moscow Engineering Physics Institute), Kashirskoe sh. 31, Moscow 115409, Russia. [7]These authors contributed equally: Jia-Wei Zhao, Kaihang Yue, Hong Zhang. ✉e-mail: ligaoren@scu.edu.cn

$IrO_x$ structure. Nevertheless, most studies overlook the impact of B-site dissolution on surface oxygen stability and Ir-O coordination structure, the precise reasons contributing to the high OER catalytic performance of Co-doped $SrIrO_3$ catalysts remain unclear. This issue primarily stems from the challenge faced by researchers in elucidating two key aspects of the OER catalytic process: (1) The key role of Co dissolution in catalytic processes and (2) the influence of surface and bulk Co on $IrO_x$ sites.

To address the two key challenges mentioned above, we designed a B-site Co-doped $SrIrO_3$ system to discern the dissolution mechanism at catalyst sites and the origins of $IrO_x$ catalytic activity, as shown in Fig. 1a. In situ inductively coupled plasma mass spectrometry (ICP-MS) experiments revealed simultaneous ion dissolution of Co and Sr, caused by acid corrosion prior to OER process. At the OER potential (1.60 V vs. RHE), the dissolution phenomenon was found to be negligible. Along with theoretical calculations, a series of in situ experiments including in situ Raman mapping, in situ XAS, and differential electrochemical mass spectrometry (DEMS) were conducted. These results highlighted the role of Co in two critical ways: (1) Surface Co reduces the stability of the Co-O-Ir bridge oxygen in $SrIrO_3$, leading to the rapid exposure of the low-coordination $IrO_x$ structure; (2) Bulk lattice Co optimizes the OOH binding energy of $IrO_x$, consequently reducing the overpotential. Thus, the synthesized Co-doped $SrIrO_3$ demonstrated high OER activity, markedly surpassing commercial $IrO_2$ catalysts in PEM water electrolyzer. The insights obtained from this research would significantly enhance the understanding of high OER catalytic performance of $SrIrO_3$-based perovskite catalysts, providing key insights for designing and preparing high-performance acidic OER catalysts.

## Results

### Fabrication and characterizations of Co-doped $SrIrO_3$ samples

$SrIrO_3$ doped with different amounts of Co were synthesized by using the sol-gel method, and were fully acid washed before use. The samples were denoted as follows: $Sr_2IrCoO_x$ ($SI_1C_1$), $Sr_3Ir_2CoO_x$ ($SI_2C_1$), $Sr_5Ir_4CoO_x$ ($SI_4C_1$), $Sr_7Ir_6CoO_x$ ($SI_6C_1$), $Sr_9Ir_8CoO_x$ ($SI_8C_1$), and $SrIrO_x$ (SI).

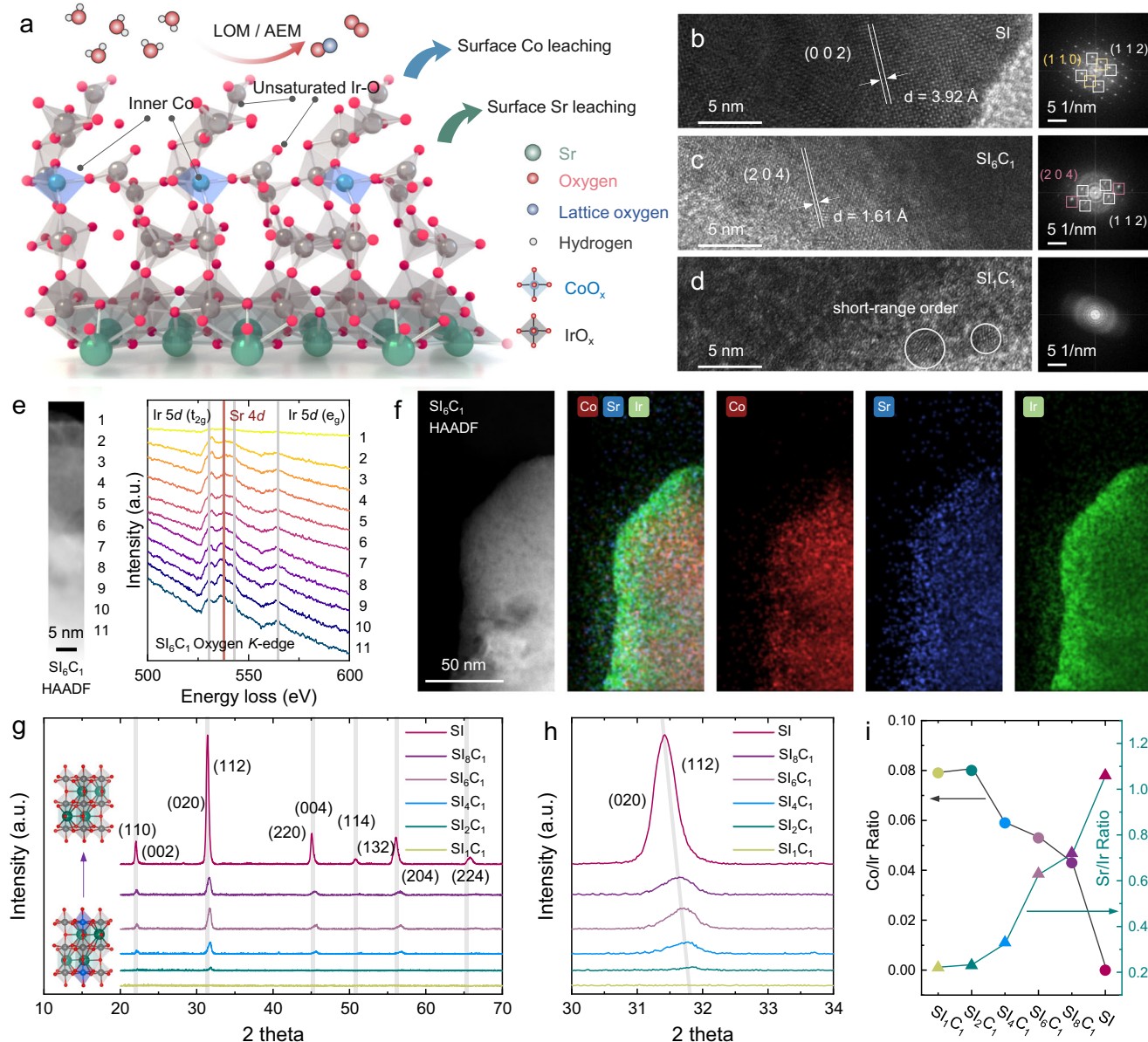

**Fig. 1 | Characterizations of samples. a** OER catalytic mechanism diagram of Co-doped $SrIrO_3$ catalyst; **b**–**d** HRTEM images and corresponding FFT diagrams of SI, $SI_6C_1$ and $SI_1C_1$; **e** O K-edge EELS spectra of $SI_6C_1$; **f** HAADF image of $SI_6C_1$ and corresponding EDS mapping images; **g, h** XRD spectra of $SI_1C_1$, $SI_2C_1$, $SI_4C_1$, $SI_6C_1$, $SI_8C_1$, and SI samples. **i** ICP–MS diagram of Co/Ir and Sr/Ir ratios of $SI_1C_1$, $SI_2C_1$, $SI_4C_1$, $SI_6C_1$, $SI_8C_1$, and SI samples.

The crystal structures of these samples were analyzed through high-resolution transmission electron microscopy (HRTEM) and X-ray diffraction (XRD) in Fig. 1b–h, respectively. For HRTEM testing, three samples (SI, $SI_6C_1$, and $SI_1C_1$) were selected. Specifically, SI exhibited an orthorhombic crystal structure. Fast Fourier transform (FFT) showed that the primary exposed crystal planes included (112), (110), and (002). Despite its low surface crystallinity, $SI_6C_1$ still clearly exposed crystal planes such as (112), (204), and (020). Meanwhile, $SI_1C_1$ displayed poor surface crystallinity with only a few areas showing short-range ordered crystallinity and no clear diffraction spots in the FFT pattern. To investigate the reason for the diminished crystallinity observed in $SI_6C_1$ sample, we conducted electron energy loss spectroscopy (EELS) analyses, the results of which are presented in Fig. 1e and Supplementary Fig. 3. The peaks at 532, 542, and 564 eV can be ascribed to the Ir 5d, while the peak at 536 eV is attributable to the Sr 4d[31,32]. Comparative analysis reveals that the Sr 4d peak of the SI sample is generally higher than that of the $SI_6C_1$ sample, suggesting that the presence of Sr contributes to maintaining the crystalline structure. Moreover, we identified a significant discrepancy between the surface and bulk Sr concentrations within the SI and $SI_6C_1$ samples, with this effect being particularly pronounced in the $SI_6C_1$ sample. This indicates that Co dissolution also impacts the proportion of Sr presenting on the surface. Further investigations were undertaken through energy-dispersive X-ray spectroscopy (EDS) mapping (Fig. 1f). The results confirm that the dissolution of Sr/Co and the subsequent formation of $IrO_x$.

XRD revealed that the diffraction peaks of SI were consistent with typical pseudocubic (Pnma) perovskites. Also, Co-doped SI also demonstrated similar diffraction peaks, confirming the successful synthesis of the perovskite-based catalyst[21,26]. As shown in Fig. 1h, Co doping significantly reduced the crystallinity of SI and resulted in the diffraction peak shifting to the high angle, indicating lattice contraction in the Co-doped SI. In particular, $SI_1C_1$ and $SI_2C_1$ displayed almost no diffraction peaks, possibly due to the substantial dissolution of surface Co and Sr by acid washing process. These findings suggest that excessive Co doping may disrupt the pseudocubic perovskite structure after acid washing, a conclusion in line with the HRTEM results. The overall composition of the samples was analyzed by using ICP-MS, as shown in Fig. 1i. Notably, there was a significant discrepancy between the Co and Ir initial ratio and the final composition ratio of the samples. Samples with poor crystallinity, specifically $SI_1C_1$ and $SI_2C_1$, exhibited the highest Co/Ir ratio, with $SI_2C_1$ slightly higher than $SI_1C_1$. This could possibly be attributed to the difficulty in maintaining the perovskite structure in $SI_1C_1$, which led to a large amount of Co dissolution during the acid washing process and, consequently, a reduced Co/Ir ratio. Despite possessing the highest Co/Ir ratios, these two samples were still significantly lower than the initial ratio, further confirming the instability of surface Co in acidic conditions. The Co/Ir ratios of $SI_4C_1$, $SI_6C_1$, and $SI_8C_1$ showed a decreasing trend from $SI_4C_1$, $SI_6C_1$ to $SI_8C_1$ as shown in. Figure 1i suggesting that lattice maintenance assists in stabilizing Co atoms in the bulk phase. Among various SI-based samples, the $SI_1C_1$ and $SI_2C_1$ displayed the lowest Sr/Ir ratio, and the Sr/Ir ratio significantly increased with the decrease of Co doping. The Sr/Ir ratio of the SI sample was slightly higher than 1, indicating a slightly higher Sr proportion compared to Ir in bulk structure, as suggested by the EELS results shown in Supplementary Fig. 3. These conclusions were further confirmed by additional characterization methods such as scanning electron microscope (SEM), X-ray photoelectron spectroscopy (XPS), and Raman spectroscopy, as shown in Supplementary Fig. 5a, b.

## High OER catalytic performance of Co-doped $SrIrO_3$

The OER catalytic performances of the $SI_1C_1$, $SI_2C_1$, $SI_4C_1$, $SI_6C_1$, $SI_8C_1$, and SI samples were investigated in acidic medium. The linear sweep voltammetry (LSV) curves for $SI_1C_1$, $SI_2C_1$, $SI_4C_1$, $SI_6C_1$, $SI_8C_1$, and SI samples were recorded after 10 cyclic voltammetry (CV) cycles, as shown in Fig. 2a. The results indicate that $SI_6C_1$ necessitates an

overpotential of only 245 mV to reach a current density of 10 mA/cm², which is approximately 5 times higher than that of SI at the same potential. Given that the OER catalytic process primarily occurs on the surface, the performance of $SI_1C_1$, $SI_2C_1$, $SI_4C_1$, $SI_6C_1$ and $SI_8C_1$ samples is generally superior and exhibits a similar trend. This aligns with previous characterization results, suggesting that despite varying Co doping ratios, the catalysts' surface structures are comparable. As presented in Fig. 2b, c, the $SI_1C_1$, $SI_2C_1$, $SI_4C_1$, $SI_6C_1$ and $SI_8C_1$ samples show Tafel slopes within the range 51.5–53.6 mV/dec, significantly lower than that of SI (59.5 mV/dec). This suggests that Co participation can enhance the reaction kinetics of OER. As shown in Fig. 2d, a comparison of the mass activity of the $SI_1C_1$, $SI_2C_1$, $SI_4C_1$, $SI_6C_1$, $SI_8C_1$, SI and $IrO_2$ samples reveals that the Co-doped samples all demonstrated high activity, significantly surpassing those of SI and $IrO_2$.

To study the potential applications of SI series catalysts in water electrolysis, we further examined the electrocatalytic performance of three samples (SI, $SI_6C_1$, and $IrO_2$) for water electrolysis by PEM. The schematic diagram of the PEM water electrolyzer is presented in Fig. 2e. The performance results of the PEM water electrolysis, shown in Fig. 2f, reveal that the catalytic activity of $SI_6C_1$ is significantly higher than those of SI and $IrO_2$. It can achieve a current density exceeding 1000 mA/cm² at 2.0 V cell voltage at 85 °C, which is much higher than those of the SI catalyst (580 mA/cm²) and $IrO_2$ catalyst (560 mA/cm²) under the same conditions. We further tested the long-term stability of $SI_6C_1$ catalyst for PEM water electrolysis as shown in Fig. 2g and Supplementary Fig. 6. The results indicate that the $SI_6C_1$ catalyst exhibits high stability, with a performance decay rate of 0.21 mV/h, which is comparable to that of the SI catalyst and commercial $IrO_2$ catalyst. This exceptional high stability indicates that the $SI_6C_1$ catalyst exhibits the potential for practical application in PEM water electrolysis.

## The roles of Co doping and dissolution in $SrIrO_3$ studied by theoretical calculations

The impact of Co doping on $SrIrO_3$ was investigated through a theoretical study. As numerous early studies had substantiated that $SrIrO_3$-based catalysts undergo significant Sr dissolution during OER[23–25]. This was examined by analyzing the Pourbaix diagrams of $SrIrO_3$ and $Sr_4Ir_3CoO_{12}$, particularly focusing on their stability in acidic medium (Fig. 3a and Supplementary Fig. 7). The findings reveal a marked similarity in the stability properties of $SrIrO_3$ and $Sr_4Ir_3CoO_{12}$. Under acidic conditions, Sr displays a thermodynamic inclination towards dissolution, thereby exposing a multitude of $IrO_x$ sites on the surface. Additionally, Co in the $Sr_4Ir_3CoO_{12}$ also exhibits instability in acidic medium, which could also result in substantial dissolution, consistent with the EDS mapping and ICP-MS results in Fig. 1.

As indicated by the Pourbaix diagram, the structure of catalyst is significantly influenced by the applied voltage. However, the two catalysts do not exhibit notable thermodynamic differences near the oxygen stability curve, although they display pronounced differences in thermodynamic tendencies across varying pH values. This suggests that the effect of $SrIrO_3$ dissolution at different OER potentials may be significantly less than the effect of electrolyte pH, particularly in strongly acidic medium. Previous research suggests that the Ir-O coordination number of surface $IrO_x$ on $SrIrO_3$ during OER catalysis is ~4.5[25]. While the presence Sr content at the trace amount has been detected in numerous studies, determining the exact state of Sr at the subnanometer scale remains a challenge[23]. By combining the results from the Pourbaix diagrams and above characterizations, seven models were constructed to investigate the theoretical OER catalytic sites and catalytic performance of $SrIrO_3$ and Co-doped $SrIrO_3$ (Fig. 3b and Supplementary Fig. 8).

The theoretical calculations were conducted on the models, as shown in Fig. 3c. The computed free energy diagrams (Fig. 3d) reveal that the rate-determining step for all models is the formation of *OOH. The Co doping, whether at the surface and bulk phase, which would

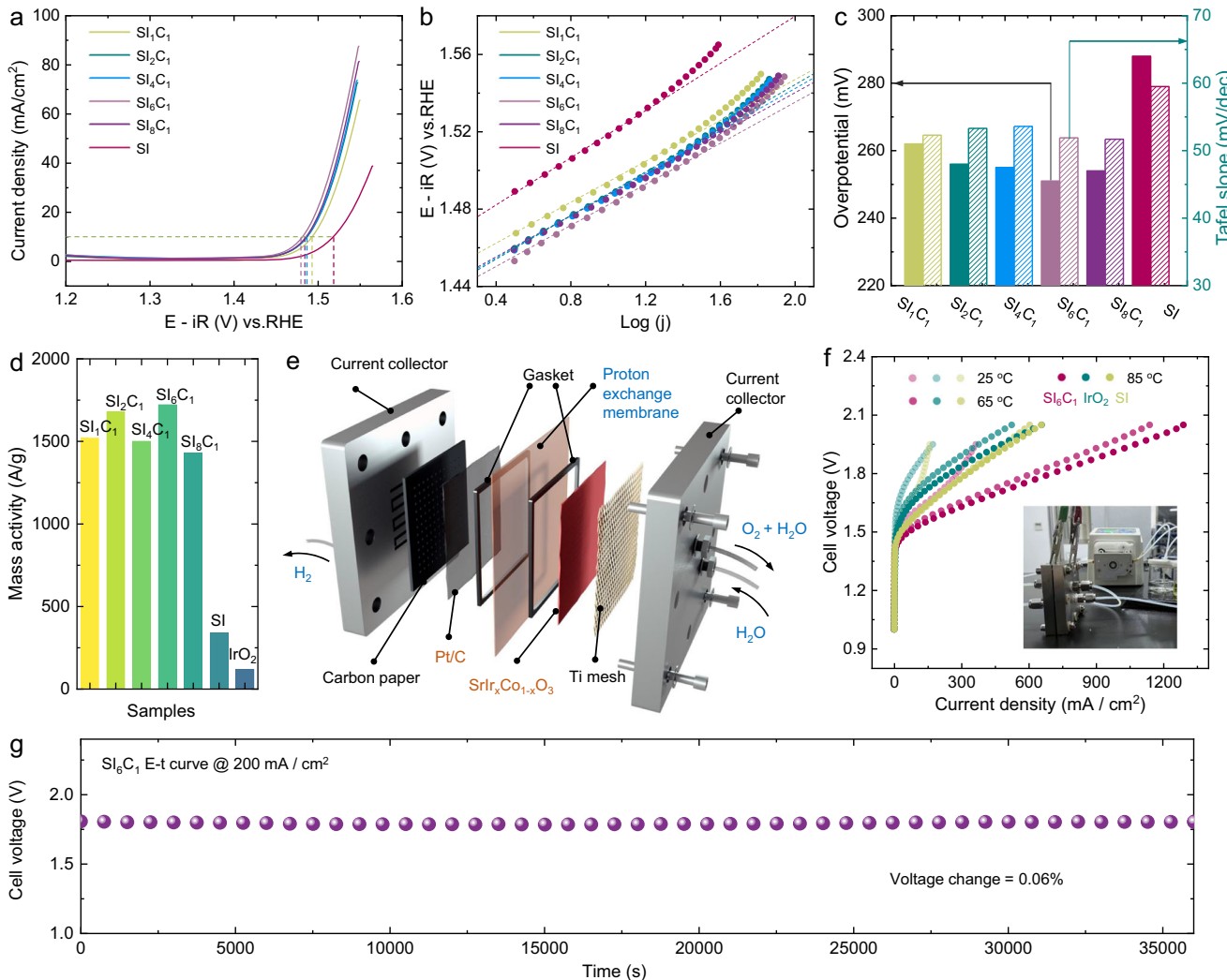

**Fig. 2 | Electrochemical measurements of samples. a** OER polarization curves of $SI_1C_1$, $SI_2C_1$, $SI_4C_1$, $SI_6C_1$, $SI_8C_1$, and SI samples with a mass loading of 0.025 mg/cm$^2$ in 0.5 M $H_2SO_4$ and (**b**) corresponding Tafel slopes. The resistance values for $SI_1C_1$, $SI_2C_1$, $SI_4C_1$, $SI_6C_1$, $SI_8C_1$, SI and $IrO_2$ were 3.9, 3.7, 3.6, 4.7, 3.2, 4.6 and 4.4 Ω, respectively. **c** Comparison of overvoltages and Tafel slopes of $SI_1C_1$, $SI_2C_1$, $SI_4C_1$, $SI_6C_1$, $SI_8C_1$, and SI samples. **d** Comparison of OER mass activity of $SI_1C_1$, $SI_2C_1$, $SI_4C_1$, $SI_6C_1$, $SI_8C_1$, SI and $IrO_2$ samples. **e** Schematic diagram of PEM water electrolysis device. **f** PEM water electrolysis performance of $SI_6C_1$, $IrO_2$, and SI samples, in set: PEM water electrolysis device photograph. **g** PEM water electrolysis stability of $SI_6C_1$ sample.

not change the coordination structure of $IrO_6$ octahedron, can only enhance OER catalytic activity to a certain extent. However, such activity enhancements are restricted, especially for bulk-doped Co, where the OER overpotential is reduced by only 15 ~ 42 mV. In contrast, the surface unsaturated $IrO_x$, which forms following Co dissolution, exhibits a significant improvement in the catalytic activity for oxygen evolution, with the overpotential decreasing by 311 ~ 344 mV (Fig. 3e). These findings suggest that Co may not directly participate in the catalysis, but rather promote the surface reconstruction through site dissolution, leading to rapid exposure of more low-coordination $IrO_x$ active sites.

To further investigate the activity origin, density of states (DOS) analyses were performed on these models (Fig. 3f). The data suggest that Co doping significantly affects the Co-O-Ir bridge oxygen, shifting the O 2*p* band center closer to the Fermi level. According to the related studies, such a shift in the O 2*p* band center substantially affect the surface stability of the catalyst and may be associated with oxygen dissolution[33,34]. This indicates that the primary effect of Co doping is to alter the surface stability of $SrIrO_3$, and promote the surface reconstruction and the formation of $IrO_x$ active sites. Additionally, the doping and dissolution of Co significantly influence the Ir 3*d* band

center (Fig. 3g). The displacement of the metal 3*d* orbitals typically directly impacts the binding energy to the adsorbate. In this study, the negative shift of the Ir 3*d* band center led to a decrease in *OOH free energy, significantly enhancing the OER catalytic activity (Fig. 3g). By establishing the relationship between the Ir 3*d* and O 2*p* band centers and the theoretical overpotential, a new OER volcano plot was formulated (Fig. 3h). The data revealed that the moderate O 2*p* band center and the lower Ir 3*d* band center play crucial roles in the catalytic activity of $SrIrO_3$-based catalysts.

### In situ characterizations of surface structural changes of catalysts

To elucidate the catalytic mechanism, DEMS tests were first conducted, as shown in Fig. 4a. The surfaces of $SI_1C_1$, $SI_2C_1$, $SI_4C_1$, $SI_6C_1$, $SI_8C_1$, SI, and $IrO_2$ samples were labeled with $^{18}O$, as shown in Supplementary Figs. 10–12. For each sample, $m/z = 34$ signals at different cyclic voltammetry cycles were collected, and the ratio of $^{18}O^{16}O$ to $^{16}O_2$ was utilized to eliminate the natural abundance of $^{18}O$ in the air. As shown in Supplementary Fig. 13, the lattice oxygen evolution reaction (LOER) trend of most catalysts across different cyclic voltammetry cycles is similar, and the Co doping ratio appears to facilitate the

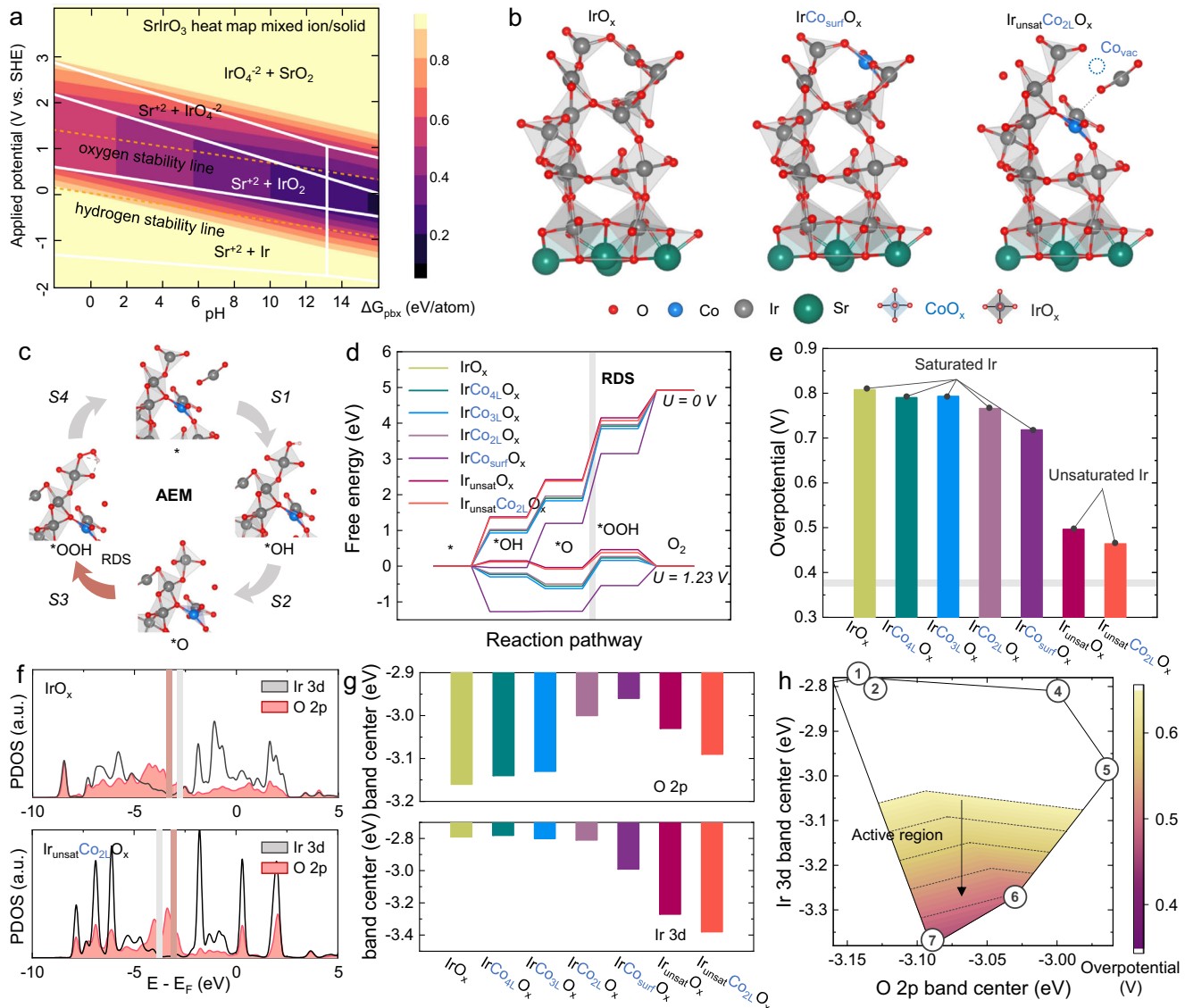

**Fig. 3 | Theoretical calculations of SrIrO₃-based perovskite catalysts. a** Pourbaix diagram of SrIrO₃. **b** Possible computational models of IrOₓ, IrCosurfOₓ (surface Co doping) and IrunsatCo2LOₓ (second layer Co doping with unsaturated IrOₓ) for DFT calculations. **c** Adsorbate evolution mechanism (AEM) diagram. **d** OER free energy diagrams of different models. **e** Overpotential of different computational models. **f** Density of states diagrams for IrOₓ and IrunsatCo2LOₓ, and (**g**, **h**) energy band center and volcano plot for different computational models.

release of lattice oxygen. The results indicated that Co doping reduces the stability of the surface oxygen in SrIrO₃, consistent with O 2*p* center calculation results. Notably, the LOER of SI₁C₁ and SI₂C₁ is the most prominent, with SI₂C₁ slightly surpassing SI₁C₁. This phenomenon has been previously explained by XRD and ICP-MS characterizations, as SI₁C₁ has difficulty in maintaining the perovskite lattice, the proportion of Co doping decreases. In contrast, SI and IrO₂ exhibit the smallest LOER process, which is linked to the stability of the catalyst surface lattice oxygen. The results indicated that Co doping and dissolution can activate the catalyst lattice oxygen, thereby accelerating the formation of IrOₓ.

In addition to LOER, in situ ICP-MS experiments were performed to detect the phenomenon of ion dissolution during OER, as shown in Fig. 4b, c. We observed that both SI₆C₁ and SI undergo considerable dissolution when immersed in an acidic electrolyte. Coupling this observation with prior XRD and Raman results, it suggests that the dissolution process is triggered by the rapid dissolution of Sr/Co-related oxides or compound heterophases. Furthermore, the subsequent steady dissolution trend indicates that ion dissolution of the

catalyst does not necessarily occur during the catalytic process. It is noteworthy that the Sr dissolution phenomenon of SI is more significant than that of SI₆C₁, which is attributed to the higher concentration of surface and bulk Sr in SI. Moreover, SI₆C₁ is also accompanied by a small amount of Co dissolution, thus further confirming that Co dissolution promotes the formation of unsaturated IrOₓ.

To confirm the structural information of the catalyst, an in situ Raman mapping study was conducted, as shown in Fig. 4d, e. The peak around 600 cm⁻¹ can be attributed to the Ir-*μ*-oxo stretching vibration of IrOₓ (involving the unprotonated bridge oxygen, Ir³⁺), and the characteristic peaks around 550 cm⁻¹ and 720 cm⁻¹ can be attributed to the typical Eg and B₂g vibrational peaks of IrO₂. In addition, the characteristic peaks around 300 and 700 cm⁻¹ can be attributed to Sr-related oxide or compound[35–38]. The figure shows that before and after the OER process, SI exhibits an obvious Ir-*μ*-oxo stretching vibration peak at 600 cm⁻¹, confirming the existence of the perovskite IrO₆ structure. However, for SI₆C₁, a significant decrease in the Ir-*μ*-oxo stretching vibration peak is observed. This may be attributed to the

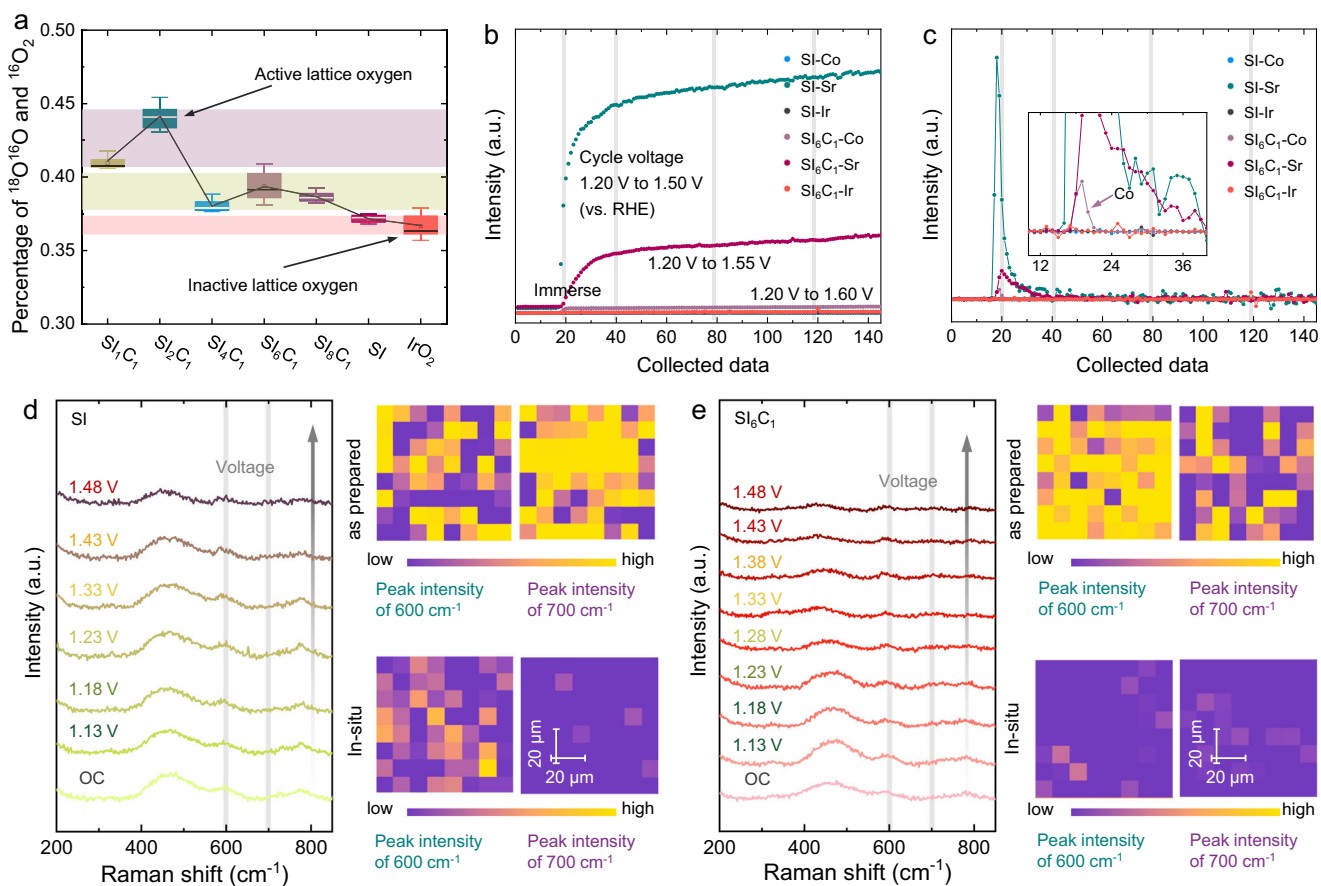

**Fig. 4 | In situ characterizations of samples. a** The $^{18}O^{16}O$ percentage of the samples test by DEMS. **b** In situ ICP-MS diagram of SI and $SI_6C_1$ and (**c**) its differential transformation diagram, in set: enlarged diagram. **d**, **e** In situ Raman and Raman mapping of SI samples and $SI_6C_1$ samples.

destruction of the $IrO_6$ structure by Codissolution to form the low-coordination $IrO_x$ structure, leading to a significant reduction in peak intensity. Moreover, the characteristic peaks of Sr-related oxides/compounds near 300 and 700 $cm^{-1}$ nearly disappear from the open circuit voltage (OC). Together with the results of in situ ICP–MS, it can be further confirmed to be caused by the dissolution of Sr-related oxides/compounds.

## Key evidences of highly active low-coordination $IrO_x$ in Co-doped $SrIrO_3$ for OER

To elucidate the formation mechanism of $IrO_x$ on the surface, comprehensive in situ EXAFS spectra was recorded at the Ir-$L_{III}$ edge to monitor the evolution of local coordination of Ir, as shown in Fig. 5a–e. Initially, we examined the coordination changes of the SI sample, as demonstrated in Fig. 5a, b. The pristine state SI sample owns a Ir-O coordination number of 5.6, indicating the existence of substantial intact Ir-O octahedral structure within the catalyst. During the pre-OER stage (OC, 1.23 V, 1.43 V vs. RHE), the Ir-O coordination numbers of SI sample exhibited a pattern of initial increase followed by a decrease.

Previous reports showed a significant augmentation in the Ir-O coordination number of $SrIrO_3$ thin film samples during OER, which was ascribed to the oxygen refilling process[24,25]. In this study, the dissolution of Sr led to alterations in the surface structure of SI. However, there was no significant LOER in SI sample. Therefore, during the OER oxidation process, the rate of $H_2O$ filling exceeded the rate of LOER, resulting in the observed trend in coordination numbers. During the OER stage (1.63 V vs. RHE), the Ir-O coordination number of the SI sample exhibited a minor increase, suggesting that at high potentials, the lattice oxygen of SI remained inactive, leading to the higher $H_2O$ filling rate than the LOER rate[25]. Additionally, at 1.23 V vs. RHE, a

significant reduction in the Ir-O bond length of the SI sample was observed, potentially indicating a transformation of the octahedral corner-sharing structure during the $H_2O$ filling process[26]. In sum, as shown in Fig. 5c, the surface site mechanism of the SI sample undergoes a dynamic process from a relatively complete perovskite structure to site dissolution and adsorbate filling, ultimately transitioning into a saturated $IrO_x$ structure.

We next analyzed the coordination changes in the $SI_6C_1$ sample, as illustrated in Fig. 5d, e. The initial Ir-O coordination number of $SI_6C_1$ was 6.0, suggesting a more perfect Ir-O octahedral structure compared with SI. However, prior to the OER stage (OC, 1.23 V, 1.43 V vs. RHE), the Ir-O coordination number of the $SI_6C_1$ sample exhibited a notable decreasing trend. This is consistent with the formation of low-coordination $IrO_x$ as observed in in situ ICP-MS results. Furthermore, during the OER stage, the Ir-O coordination number of $SI_6C_1$ further decreased to 5.4, significantly lower than SI, confirming that the active lattice oxygen in $SI_6C_1$ further facilitated the formation of highly active low-coordination $IrO_x$.

To validate the generation of the unsaturated $IrO_x$ structure, in situ XANES spectra were analyzed at the Ir-$L_{III}$ edge to study the oxidation state of Ir, as shown in Fig. 5g, h. The Ir-$L_{III}$ absorption edge of SI displayed an increasing white line intensity during the OER process, indicating an increase in the oxidation state of Ir. Differently, the Ir-$L_{III}$ absorption edge of $SI_6C_1$ exhibited a slight decrease in white line intensity, signifying a reduction in the oxidation state of Ir. Combined with the valence state information of reference samples Ir and $IrO_2$, the mechanism of Ir valence state change during OER can be inferred. During the OER process of SI, the dissolution of Sr and the filling of $H_2O$ led to the saturation of the Ir-O coordination, displaying a higher valence state. Conversely, for $SI_6C_1$, the dissolution of Co resulted in

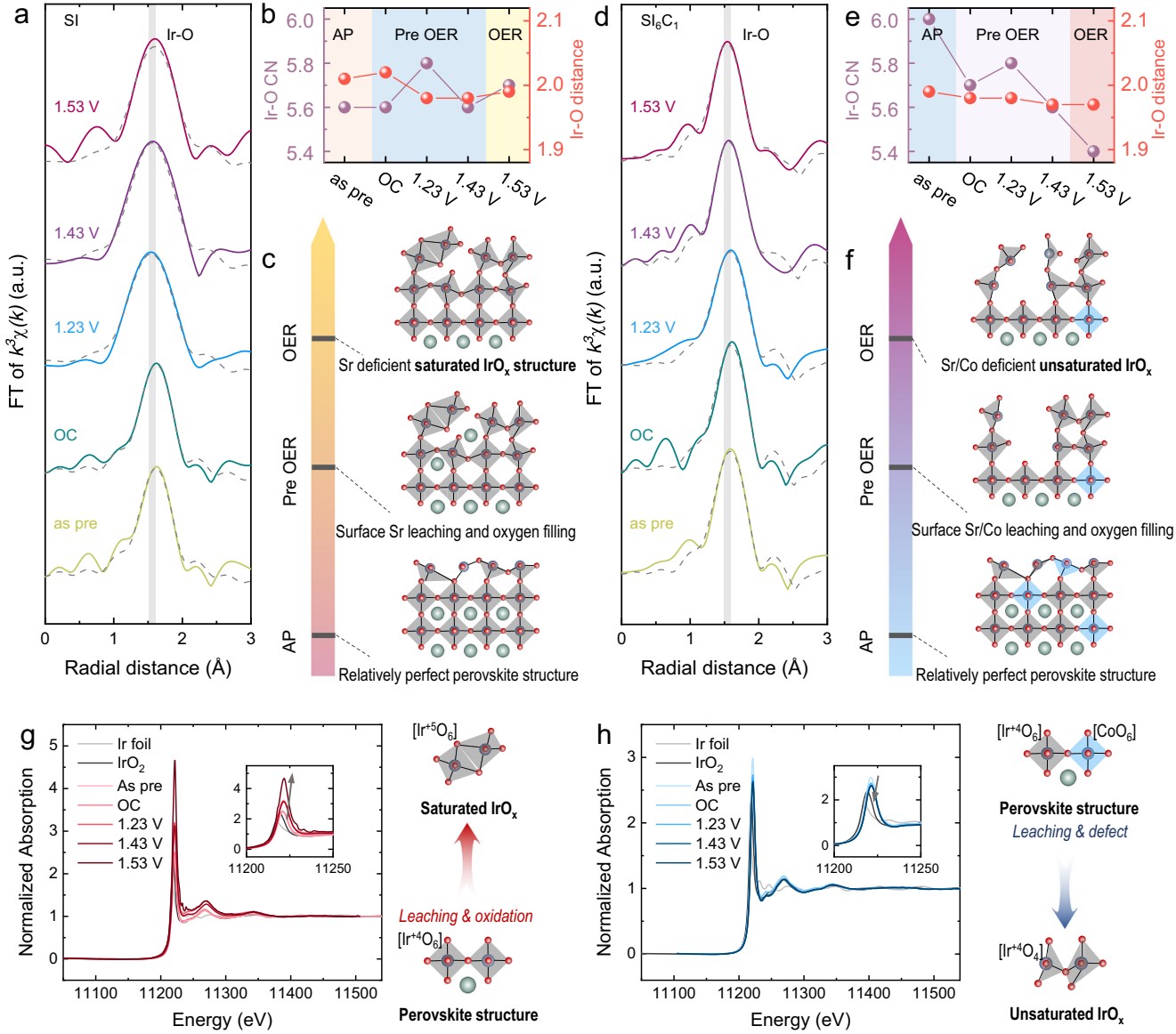

**Fig. 5 | In situ XAS characterization of samples. a** R-space fitting diagram for SI sample. **b** Changes in Ir-O coordination and Ir-O bond length of SI sample. **c** Structure change diagram of SI sample; **d** R-space fitting diagram of SI$_6$C$_1$ sample. **e** Ir-O coordination and Ir-O bond length variation diagram of SI$_6$C$_1$ sample.

**f** Structure change diagram of SI$_6$C$_1$ sample. **g** Ir−$L_{III}$ Absorption edge diagram of SI sample, in set: enlarged diagram. **h** Ir−$L_{III}$ absorption edge diagram of SI$_6$C$_1$ sample, in set: enlarged diagram.

the formation of unsaturated Ir-O, leading to a decrease in the average Ir valence state. This observation further substantiates that Co can enhance the generation of unsaturated IrO$_x$ structures, aligning with the results from prior characterizations.

## Discussion

### OER catalytic mechanism on Co-doped SrIrO$_3$ catalyst

Our study has elucidated the surface reconstruction process of SIC series catalysts by theoretical calculations and a comprehensive series of in situ characterizations. We now proceed to discuss and summarize the potential OER mechanisms inherent to SIC series catalysts. There are two widely recognized mechanisms for OER, specifically the adsorbate evolution mechanism (AEM) and the lattice oxygen mechanism (LOM), as shown in Fig. 6a, b[39,40]. Our theoretical calculations have demonstrated that the surface O 2$p$ band center of the Co/Ir model is closer to the Fermi level compared to the Ir model, which suggests that the bridging oxygen of Co-O-Ir is thermodynamically predisposed to be oxidized[40]. In situ Raman spectroscopy revealed

that the Ir-$\mu$-oxo stretching vibration peak (indicative of bridging oxygen) in the Co/Ir system catalyst was notably reduced during the OC and OER processes when compared to the Ir system. Further, DEMS tests have found that a higher content of Co promotes the LOM.

Despite these findings, our DEMS results suggest that the AEM remains the dominant mechanism, while the LOM significantly contributes to the formation of unsaturated IrO$_x$. Consequently, we propose a distinctive catalytic mechanism, the lattice oxygen promoted adsorbate evolution mechanism (LOPAEM), as illustrated in Fig. 6c. Unlike the conventional catalytic mechanisms of OER, LOPAEM involves the synergistic action of both mechanisms. Specifically, when the oxidation rate of lattice oxygen in certain catalysts becomes excessively fast, it will lead to the formation of a large number of surface oxygen vacancies (i.e., unsaturated metal sites), thereby altering the original catalytic sites of the catalyst. These unsaturated metal sites may exhibit more efficient AEM performance, thus achieving LOPAEM. Our findings suggest that the LOM in Co/Ir system catalysts is an integral step for catalyst activation. The formation of Ir-

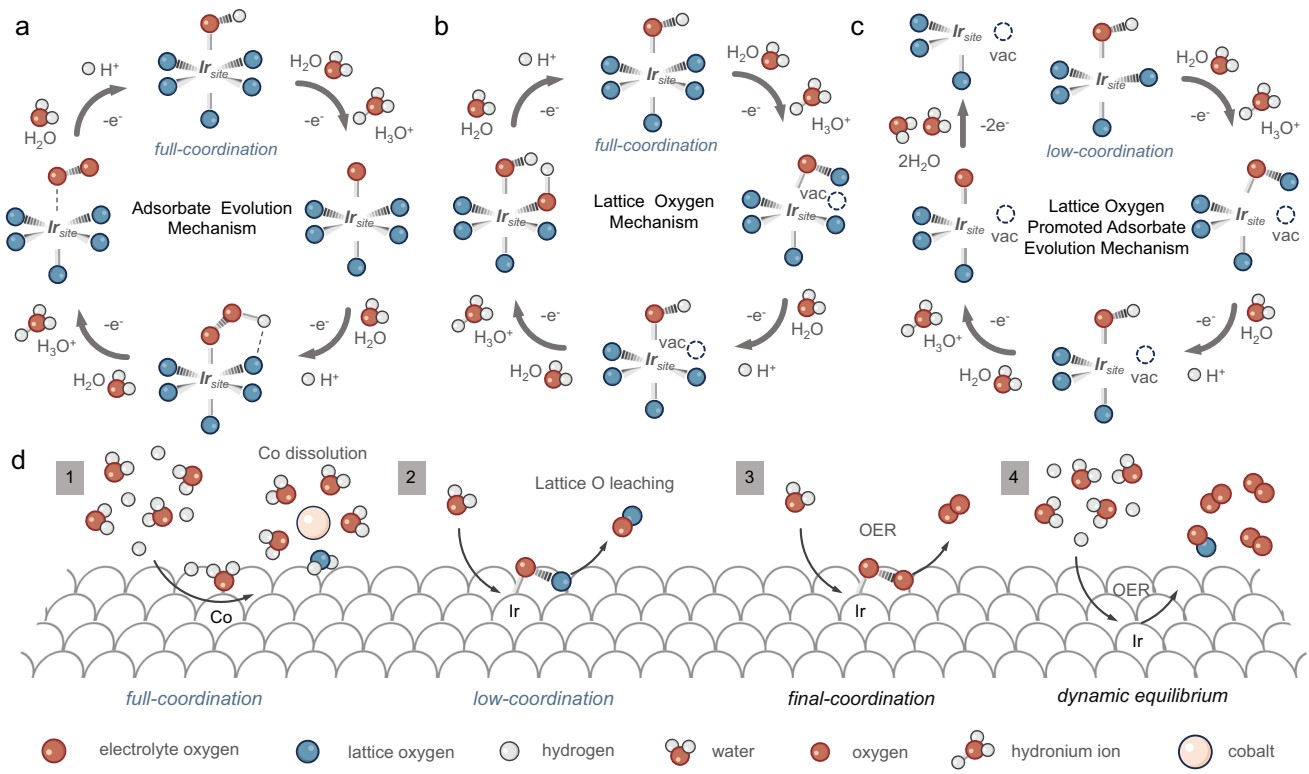

**Fig. 6 | Proposed possible OER mechanism. a** Adsorbate evolution mechanism. **b** Lattice oxygen mechanism. **c** Lattice oxygen promoted adsorbate evolution mechanism. **d** Catalytic mechanism of Co-doped SrIrO$_3$ catalyst.

O-Co$_{vac}$ substantially reduces the catalyst's AEM overpotential, thereby making it thermodynamically more favorable. This is in agreement with our theoretical calculations and DEMS results. Moreover, the LOM process is unable to produce a lower Ir-O coordination structure as the dissolution of lattice oxygen shifts the O 2$p$ center away from the Fermi level, precluding further oxidation (as detailed in the theoretical calculation section).

In light of the above findings, we have summarized the mechanism for the Co/Ir system catalyst as represented in Fig. 6d. Initially, Co on the catalyst's surface dissolves under acidic conditions, removing a portion of the bridging lattice oxygen to form an IrO$_x$ structure with lower coordination. Subsequently, lattice oxygen dissolves during the OER process, forming an unsaturated IrO$_x$ structure that exhibits high AEM activity. Lastly, under the dynamic balance of adsorbate filling and LOM, the catalyst conducts AEM to efficiently facilitate the OER.

In summary, we designed a highly active catalyst through Co doping and dynamic dissolution of Co/Sr bimetallic ions to study the catalytic mechanism of SrIrO$_3$-based perovskite. Theoretical calculations and in situ characterizations (DEMS, in situ Raman mapping, in situ XAS and in situ ICP-MS) show that dynamic dissolution of Co is crucial for forming highly active unsaturated IrO$_x$. The as-synthesized catalysts exhibit higher OER reaction kinetics than SrIrO$_3$ and commercial IrO$_2$ catalyst in both electrolyzer and PEM water electrolyzer, revealing the mechanism of catalytic activity enhancement by tuning catalytic sites. This work is of great significance for understanding the high OER catalytic performance of Ir-based catalysts, and will provide an important basis for the design and preparation of high-performance acidic OER catalysts.

## Methods
### Chemicals
The chemical reagents utilized in this study were all received from the manufacturer. Potassium hexachloroiridate (IV) [K$_2$IrCl$_6$, AR, Macklin], cobalt(III) nitrate hexahydrate [Co(NO$_3$)$_3$·6H$_2$O, AR, Sigma-Aldrich],

strontium(II) nitrate [Sr(NO$_3$)$_2$, AR, Guangdong chemical reagent)], citric acid monohydrate (AR, LookChem.) were utilized as precursors.

**Preparation of SI Catalyst.** Solution A was prepared by dissolving Sr(NO$_3$)$_2$ (280 mg) and citric acid (840 mg) in 5.0 mL of deionized water. Solution B was prepared by dissolving K$_2$IrCl$_6$ (80 mg) in 4.0 mL of ethylene glycol. Solution A was then added dropwise with stirring to solution B. The resulting mixture was dried at 150 °C for 12 h to obtain a brown solid product as a precursor. Subsequently, the precursor was calcined in air at 200 °C for 6 h, 300 °C for 6 h, 500 °C for 3 h, and 700 °C for 6 h with a heating rate of 2 °C/min. Afterward, the excess SrCO$_3$ impurities were removed by reacting with a 1.0 M HCl solution for 12 h to obtain SrIrO$_3$ (SI) catalyst.

**Preparation of SI$_1$C$_1$ catalyst.** Solution A was prepared by dissolving Sr(NO$_3$)$_2$ (280 mg) and citric acid (840 mg) in 5.0 mL of deionized water. Solution B was prepared by dissolving K$_2$IrCl$_6$ (40 mg) and Co(NO$_3$)$_2$ (24 mg) in 4.0 mL of ethylene glycol. The subsequent steps were identical to the preparation of the SI catalyst described above.

**Preparation of SI$_2$C$_1$ catalyst.** Solution A was prepared by dissolving Sr(NO$_3$)$_2$ (280 mg) and citric acid (840 mg) in 5.0 mL of deionized water. Solution B was prepared by dissolving K$_2$IrCl$_6$ (53 mg) and Co(NO$_3$)$_2$ (16 mg) in 4.0 mL of ethylene glycol. The subsequent steps were identical to the preparation of the SI catalyst described above.

**Preparation of SI$_4$C$_1$ catalyst.** Solution A was prepared by dissolving Sr(NO$_3$)$_2$ (280 mg) and citric acid (840 mg) in 5.0 mL of deionized water. Solution B was prepared by dissolving K$_2$IrCl$_6$ (64 mg) and Co(NO$_3$)$_2$ (10 mg) in 4.0 mL of ethylene glycol. The subsequent steps were identical to the preparation of the SI catalyst described above.

**Preparation of SI$_6$C$_1$ catalyst.** Solution A was prepared by dissolving Sr(NO$_3$)$_2$ (280 mg) and citric acid (840 mg) in 5.0 mL of deionized water. Solution B was prepared by dissolving K$_2$IrCl$_6$ (68 mg) and Co(NO$_3$)$_2$ (7 mg) in 4.0 mL of ethylene glycol. The subsequent steps were identical to the preparation of the SI catalyst described above.

**Preparation of SI$_8$C$_1$ catalyst.** Solution A was prepared by dissolving Sr(NO$_3$)$_2$ (280 mg) and citric acid (840 mg) in 5.0 mL of deionized water. Solution B was prepared by dissolving K$_2$IrCl$_6$ (71 mg) and Co(NO$_3$)$_2$ (5 mg) in 4.0 mL of ethylene glycol. The subsequent steps were identical to the preparation of the SI catalyst described above.

**Materials characterizations.** Characterization of the atomic-level crystal structure was performed using an aberration-corrected scanning transmission electron microscope (JEM-ARM200P, JAPAN) operated at 300 kV. Energy-dispersive X-ray (EDX) analysis was used to measure the relative elemental content. X-ray diffraction (XRD) patterns of SrIrO$_3$ were recorded on an X-ray diffractometer (Smart lab) using Cu-Kα radiation ($\lambda = 1.5418$ Å) with a step size of 0.02° and a step time of 0.2 s in the 20°–80° range. X-ray photoelectron spectroscopy (XPS) was performed using a Thermo Scientific K-Alpha X-ray photoelectron spectrometer, and all XPS spectra were calibrated using the C 1$s$ line at 284.8 eV. The surface morphology of SrIrO$_3$ was characterized using a scanning electron microscope (TESCAN MIRA LMS). Considering the high acid resistance of SrIrO$_3$, anti aqua regia was prepared by mixing hydrochloric acid and nitric acid in a 1:3 ratio for the experiment. In this solution, 1.0 mg of SrIrO$_3$ powder was dissolved in 10 mL of aqua regia and left to stand for 1–3 week after thorough ultrasonic treatment. Finally, the proportions of each element in SrIrO$_3$ were determined by ICP-MS (iCAP RQ) analysis.

**In situ characterizations and LOER confirmation.** X-ray absorption spectra of Ir $L$-edge were obtained at the BL17B and BL20U1 beamline of the Shanghai Synchrotron Radiation Facility (SSRF). The spectra were collected either in transmission mode or fluorescence mode using a Lytle detector. The corresponding reference samples were collected in transmission mode. The samples were ground and uniformly applied to special adhesive tape. In situ XANES characterizations were performed in fluorescence mode at the same beamline. The samples were sprayed onto carbon paper at a loading of 1.0 mg cm$^{-2}$ as the working electrode. The measurements were conducted under the same conditions as the OER measurements in a self-designed cell. The in situ Raman spectroscopy characterizations were carried out using an inVia confocal Raman microscope from Renishaw. The laser power was set at 532 nm with 1% power at a grating of 1800 mm/1, the silicon peak was calibrated before testing. During the OER process, differential electrochemical mass spectrometry (DEMS) measurements were conducted using the QAS 100 apparatus from Shanghai Linglu Instruments to determine the volatile reaction products of the SI series catalysts and IrO$_2$ catalyst labeled with $^{18}$O. Saturated Ag/AgCl and Pt wires served as the reference electrode (RE) and counter electrode (CE), respectively. The working electrode (WE) was prepared by sputtering Au onto a 50 μm-thick porous PTFE membrane, followed by depositing 10 μL of catalyst ink (1.0 mg/mL) onto the Au surface. The catalyst isotopic labeling was achieved by cycling the electrode in H$_2$$^{18}$O for eight cycles using cyclic voltammetry (CV) with a scan rate of 5 mV/s in the range of 0–0.5 V vs. Ag/AgCl. Subsequently, the $^{18}$O-labeled electrode was rinsed with H$_2$$^{16}$O to remove residual H$_2$$^{18}$O. Finally, the electrode was electrochemically tested against Ag/AgCl in a 1.0 M H$_2$SO$_4$ solution at different potentials with a scan rate of 5 mV/s. The DEMS signal was normalized by current density (A/g). Simultaneously, real-time measurements of gas products with different molecular weights generated during the OER process were conducted using mass spectrometry. In situ ICP-MS experiments were performed using a Thermo Scientific iCAP RQ instrument. The experimental setup

consisted of a standard three-electrode cell with a 3 mm glassy carbon working electrode, consistent with the electrochemical testing. The reference electrode used was a saturated calomel electrode (Hg/HgCl$_2$), and the counter electrode was a platinum foil electrode. To ensure accurate detection of ion distribution, a stirrer was employed to prevent leaching and dissolution errors, with data sampling occurring every 15 s.

**Electrochemical characterizations.** To prepare the catalyst ink, a 0.5 mg amount of catalyst was mixed with 1.0 ml of a 0.05 wt% Nafion solution and neutralized. Subsequently, a 10 μl volume of the prepared ink was deposited onto a glassy carbon electrode (GCE) with a diameter of 5 mm and dried using an infrared lamp. Prior to use, the GCE was polished with 0.05 μm alumina powder and rinsed three times with a mixture of high purity water and ethanol. Electrochemical measurements were conducted in a three-electrode system using an electrochemical workstation (CHI 760E). The reference electrode used was an Hg/HgCl$_2$ electrode in a 0.5 M H$_2$SO$_4$ electrolyte, while a carbon rod served as the counter electrode. The working electrode was the GCE with the catalyst. LSV and CV was performed in 0.5 M H$_2$SO$_4$ solution at a scan rate of 10 mV/s. A home-made PEM water electrolysis cell with a proton exchange membrane was used to evaluate the performance of SI series catalyst. The preparation step of catalyst inks is the same as above method. The total catalyst loading on the electrode was 1.0 mg and all the catalyst inks were deposited on carbon paper (1 cm × 1 cm). The cell temperature (25, 65, and 85 °C) was maintained by an electric heating plate and measured by a temperature probe in electrolyte.

**Computational methods.** Spin-polarized density functional theory (DFT) calculations were performed in the plane wave and ultrasoft pseudopotential (USPP) with Perdew-Burke-Ernzerhof (PBE) exchange functional correction as implemented in Quantum ESPRESSO[41,42]. An energy cutoff of 25 Ry was employed for the plane wave expansion of the electronic wavefunction. The atomic structures of the models were fully relaxed until self-consistency was achieved with a convergence criteria of 10$^{-6}$ Ry for the energy and 10$^{-3}$ Ry/Bohr for the atomic coordinates. To prevent interaction between layers, a vacuum slab of 12 Å was used to isolate the surface. For bulk geometry optimization, a 3 × 3 × 1 Monkhorst-Pack k-point set was used, while a 5 × 5 × 1 set was used for electronic structure calculations. The correction for every adsorbate and surface, with typical values of +0.35 eV, +0.05 eV, +0.35 eV for *OH, *O and *OOH respectively. To simulate the complex unsaturated IrO$_x$ structure, we start by constructing and optimizing the slab structure of the original SrIrO$_3$. Then, the surface Sr atoms were removed from the SrIrO$_3$ slab for further optimization. According to a previous report[25], the Ir-O coordination number on the SrIrO$_3$ surface is approximately 4.5, achieved by removing 4 Sr atoms and their corresponding 3 neighboring O atoms. The same procedure can be applied to the Co-doped system, where removing a Co atom also removes 2 neighboring O atoms.

**XAS analysis.** The acquired extended X-ray absorption fine structure (EXAFS) data were processed following standard procedures using the ATHENA module of the Demeter software package[43]. The EXAFS spectra were obtained by subtracting the post-edge background from the overall absorption and then normalized with respect to the edge-jump step. The χ(k) data were Fourier transformed to real (R) space using a Hanning window (dk = 1.0 Å$^{-1}$) to separate the contributions from different coordination shells. To determine the quantitative structural parameters around the central atoms, least-squares curve parameter fitting was performed using the ARTEMIS module of the Demeter software package.

## Data availability

The data generated in this study are provided in the Supplementary Information and are available from the authors upon request. Source data are provided with this paper.

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

## Acknowledgements

This work was supported by NSFC (52373215), Sichuan Science and Technology Program (2023NSFSC0086), and Fundamental Research Funds for the Central Universities (YJ2021156). We also thank BL17B1 and BL20U1 station at Shanghai Synchrotron Radiation Facility (SSRF) for the help in characterizations.

## Author contributions

Experiments were conceived and designed by G.-R.L. and J.-W.Z., with inputs from all authors. S.W., J.Z., D.W., J.C. and V.Y.F. performed the electrochemistry and characterization experiments. K.Y. contributed to the in situ XAS studies and PEM test. H.Z. provided help in EELS and HAADF characterizations.

## Competing interests

The authors declare no competing interests.
