## [Peer Review File · Nature Communications]

The Formation of Unsaturated IrO_x in SrIrO₃-based Perovskites by Cobalt-doping for Efficient Acidic Oxygen Evolution ReactionREVIEWER COMMENTS

Reviewer #1 (Remarks to the Author):

In this manuscript, the authors elucidated the origin of catalytic activity of SrIrO₃-based perovskite for the OER in acid. However, there are many SrIrO₃-based perovskite catalysts reported for the OER in acid medium, such as "Science 2016, 353, 1011-1014; Science advances, 2021, 7, eabc7323; nature communication, 2018, 9, 5236." They already studied the origin of catalytic activity and determined the active species IrO_x. The authors explain the Co dissolution and Co in bulk synergistically promote the activity, which fast the IrO_x formation, in fact, the IrO_x had been formed for many MIrO₃ at the beginning of OER, the work is hard to explain why the PEM test is stable even the destruction of the IrO₆ structure by Co de-doping to form the low-coordination IrO_x structure. I don't think it is suitable for the publication in Nat Comm. There are still some issues as followed:

1. In Line 114 of page 6, the authors think "The Sr/Ir ratio of SI sample was slightly higher than 1, which may be due to the acid resistance of the IrO_x structure that prevents complete dissolution even after prolonged immersion in aqua regia".

I think it is wrong, because the dissolution of Sr in SI (SrIrO₃) is much more than that of Ir with acid washing, Sr/Ir ratio is impossible higher than 1 after acid washing, the authors need try to dissolve all samples when they test ICP.

2. In general, the activity is enhanced will lead to a decreasing stability, did the authors test the stability of catalysts in solution of three-electrode system? It is stable or unstable compare to commercial IrO₂? I guess that Co-doped SrIrO₃ will not too stable in in solution of three-electrode system. Why it will extend a few hundred hours in PEM test?

3. There are errors in the annotations of Figures 1, S10, S11, S14, S15, S16, and S17.

In Figure 1, the d is missing. In Figure S10 and 11, d, e, f is missing. In S14, S15, S16 c, d is missing. In Figure S17, the c, d, e, f is missing.

4. The authors should compare the stability of SI for PEM with the prepared catalysts.

Reviewer #2 (Remarks to the Author):

This manuscript presents a study on electrocatalytic water splitting, focusing on a Co-doped SrIrO₃ system for enhancing the oxygen evolution reaction (OER) in an acidic medium. The study looks into how SrIrO₃-based perovskite-type catalysts get their high catalytic activity. This is an important part of making

hydrogen in a way that does not harm the environment. The authors have employed a combination of in-situ experiments and theoretical calculations to elucidate the role of Co doping in activating surface oxygen and optimizing adsorbate binding energy. The findings significantly contribute to our understanding of the catalytic mechanisms in acidic OER and offer valuable insights for designing efficient electrocatalysts. However, the study also presents certain limitations and areas that require clarification to bolster the presented conclusions.

1. The authors should fix the problem with the Sr:Ir ratio in SI6C1 and SI samples that was pointed out in Line 84. There is a difference between the EELS results and the compound ratios. Clarification on this matter is crucial, as it directly impacts the understanding of the catalyst composition and its subsequent catalytic behavior.

2. The absence of peaks in the XRD pattern of sample SI1C1, as shown in Figure 1g, raises questions about the phase of the material. The authors should address whether the consistent doping ratio of Ir and Co indeed leads to an amorphous phase, and if so, discuss the implications of this phase on the catalyst's performance.

3. In the computational section, it is essential for the authors to provide details on the catalyst structure built for the SI6C1 system. Information about the dissolved proportions of Sr, Co, and O in the model is needed. Additionally, the authors should explain how surface reconstruction, promoted by Co dissolution as mentioned in lines 175 and 189, is incorporated in the simulation model.

4. Regarding Line 297, a clearer explanation of the main differences and advantages of the newly proposed OER mechanism (LOPAEM) compared to traditional mechanisms (AEM and LOM) is necessary. This clarification will help in understanding the novelty and significance of the proposed mechanism in the broader context of OER research.

Reviewer #3 (Remarks to the Author):

This study reveals the origin of Co doping in SrIrO₃ for acidic OER, which results in surface unsaturated IrO_x. Although some literatures have already reported the application and the dissolution of Co-doped

SrIrO₃, this study provides a deeper analysis and understanding. Objectively speaking, its OER activity, such as the overpotential at 10 mA cm⁻², is moderate, especially compared to other recently-reported catalysts. There are several issues still needed to be addressed before publication. I suggest that the current manuscript requires a major revision.

1. The introduction of Co results in surface unsaturated IrO_x. Whether the surface unsaturated IrO_x remains stable during acidic OER operation. The material degradation is also crucial for the commercial application. Please provide the TEM image after long-term operation.
2. During DFT calculation, how are the representative models constructed. There are many possibilities for the surface unsaturated IrO_x with Co and Sr leaching.
3. In the case of in-situ Raman, are there any other key Ir-related reaction intermediates observed?
4. In fig3d, the colors of the curves are difficult to distinguish.

Response Letter

Dear Editor Ruben Rizo and Reviewers,

Thank you very much for your kind handling or reviewing our paper entitled “The Formation of Unsaturated IrO_x in SrIrO_3 -based Perovskites by Cobalt-Doping for Efficient Acidic Oxygen Evolution Reaction” (Manuscript ID: NCOMMS-23-54524-T). We sincerely thank you for providing us with the opportunity to revise our paper, and we are grateful to the reviewers for providing constructive comments. Upon thorough review of these comments, we found them to be valuable in improving our work and guiding our future research. Accordingly, we carefully considered all the comments and made meticulous revisions to our manuscript, incorporating additional experimental data such as XAS, EELS, HRTEM, DFT calculations, PEMWE, and ICP-MS. Furthermore, we have added some discussions to the manuscript and created an animation illustrating the catalytic mechanism, thereby emphasizing the central aspects of our research. The revisions are highlighted in the revised manuscript.

Here we have provided the point-by-point responses to reviewers’ comments. The list of changes and our responses to three reviewers’ comments are shown as follows. Thank you very much.

Sincerely Yours,

Gao-Ren Li

Reply to the Comments of Reviewer #1:

General comments: In this manuscript, the authors elucidated the origin of catalytic activity of SrIrO_3 -based perovskite for the OER in acid. however, there are many SrIrO_3 -based perovskite catalysts were reported for the OER in acid medium, such as "Science 2016, 353, 1011-1014; Science advances, 2021, 7, eabc7323; nature communication, 2018, 9, 5236." They already studied the origin of catalytic activity and determined the active species IrO_x . The authors explain the Co dissolution and Co in bulk synergistically promote the activity, which fast the IrO_x formation, in fact, the IrO_x had been formed for many MIrO_3 at the beginning of OER, the work is hard to explain why the PEM test is stable even the destruction of the IrO_6 structure by Co de-doping to form the low-coordination IrO_x structure. I don't think it is suitable for the publication in Nat Comm. There are still some issues as followed.

Response: We thank Reviewer #1 for the excellent comments on our manuscript. Your professional evaluations are constructive

for improving this manuscript. After reading your comments carefully, we review both of the logical issues and the innovation to further improve the quality of our manuscript. Accordingly, we have made substantial revisions in this manuscript to fully address your concerns by adding new experiments and discussions. We also feel your critical comments are highly valuable and important for us to improve the scientific rigor of our work and make the evidences more clearly to the readers.

For your general comment of *“the authors elucidated the origin of catalytic activity of SrIrO₃-based perovskite for the OER in acid. however, there are many SrIrO₃-based perovskite catalysts were reported for the OER in acid medium, such as Science 2016, 353, 1011-1014; Science advances, 2021, 7, eabc7323; nature communication, 2018, 9, 5236. They already studied the origin of catalytic activity and determined the active species IrO_x”*, our response is as follows: Recently, although there are some papers about SrIrO₃-based perovskite catalysts for the OER in acid medium, our paper has many innovative points that differ from previous papers. To highlight the innovation of our manuscript, we have provided a detailed manuscript framework to give you a clearer understanding of the ideas and innovations of our manuscript, as shown in **Figure R1**. In this paper, we designed and fabricated efficient Co-doped SrIrO₃-based perovskite catalysts with unsaturated IrO_x through site dissolution on the catalyst surface. In addition, we established a link between OER and the dynamic surface structure change of Co-doped SrIrO₃ catalysts, as well as the corresponding OER catalytic mechanism. We also analyzed the references you mentioned above

Fig. R1 The framework of our manuscript.

in detail as shown in **Figure R2**. Actually, these mentioned papers only investigated the initial and final structure of SrIrO₃ and the correlation of OER (*Science*, 2016, 353, 1011; *Sci. Adv.*, 2021, 7, eabc7323; *Nature Commun.*, 2018, 9, 5236.). However, to our best knowledge, there is no study about the dynamic changes of Co-SrIrO₃ surface structure during OER process by in-situ characterizations and DFT calculations. Significant innovations of our manuscript are listed as follows: (i) We firstly established a link between OER and the dynamic surface structure change of Co-doped SrIrO₃ perovskite catalysts; (ii) We solved the origin issues of Co during the catalytic processes of Co-doped SrIrO₃ by theoretical calculations and state-of-the-art in-situ characterizations, and proposes the lattice oxygen promoted adsorbate evolution mechanism (LOPAEM) for OER. (iii) We synthesized highly efficient Co-doped SrIrO₃ electrocatalysts for acidic OER. This work also has important reference significance for the study of catalytic reaction kinetics.

Fig. R2 The details of the SrIrO₃ related-works and the novelty of our work.

For your general comment of “*The authors explain the Co dissolution and Co in bulk synergistically promote the activity, which fast the IrO_x formation, in fact, the IrO_x had been formed for many M₁IrO₃ at the beginning of OER*”, our response is as follows: Thanks for your professional comment. Although there are some studies reporting the formation of IrO_x at the beginning of OER in many M₁IrO₃ catalysts (*Science*, 2016, 353, 1011; *Sci. Adv.*, 2021, 7, eabc7323; *Nat. Commun.*, 2018, 9, 5236; *Energy Environ. Sci.*, 2023, 16, 513), these studies did not investigate the specific active structure of IrO_x that is crucial for the catalytic activity of OER, such as the saturated IrO_x or unsaturated IrO_x. Our study demonstrates that the unsaturated IrO_x is the key to achieving high OER catalytic activity and discovers that Co effectively promotes the formation of unsaturated IrO_x. Furthermore, we elucidate the OER catalytic mechanism of unsaturated IrO_x in this paper. For ease of understanding, we created an academic animation to visually illustrate the OER catalytic mechanism of unsaturated IrO_x, and the animation is included as Supplementary Movie.

For your general comment of “*the work is hard to explain why the PEM test is stable even the destruction of the IrO₆ structure by Co de-doping to form the low-coordination IrO_x structure*”, our response is as follows, our response is as follows: Thanks for your professional comment. The catalytic performance and stability of catalysts do not necessarily have a direct relationship. Many reports on metal-doped systems indicate that the dissolution of metal elements does not necessarily lead to a decrease in catalytic performance and stability (*Nat. Commun.*, 2019, 10, 572; *Angew. Chem. Int. Ed.*, 2023, 62, e202311606; *Nat. Catal.*, 2019, 2, 763-772; *Angew. Chem.*, 2019, 131, 4619). Furthermore, in a recent study (published two months ago, *Angew. Chem. Int. Ed.*, 2023, e202313954), the commercial IrO₂ was treated with plasma, leading to the formation of low-coordinated IrO_x catalyst. The low-coordinated IrO_x catalyst shows high catalytic activity and exceptional stability, even surpassing 1000 hours for PEMWE. In our study, we believe that the LOPAEM is a crucial reason contributing to the catalytic stability of Sr₇Ir₆CoO_x (SI₆C₁) catalyst, resulting in higher stability than traditional LOM catalysts. Additionally, we have conducted numerous PEMWE experiments and found that the stability of the sample Sr₇Ir₆CoO_x (SI₆C₁) was comparable to those of the SrIrO_x (SI) and Sr₉Ir₈CoO_x (SI₈C₁) samples, and significantly higher than that of the sample Sr₂IrCoO_x (SI₁C₁). We also performed the stability analyses using a three-electrode system, and the high long-term stability was obtained. The detailed responses can be found in the following responses to the comment 2.

Comment 1: In Line 114 of page 6, the authors think “The Sr/Ir ratio of SI sample was slightly higher than 1, which may be due to the acid resistance of the IrO_x structure that prevents complete dissolution even after prolonged immersion in aqua regia”. I think it is wrong, because the dissolution of Sr in SI (SrIrO₃) is much more than that of Ir with acid washing, Sr/Ir ratio is impossible higher than 1 after acid washing, the authors need try to dissolve all samples when they test ICP-MS.

Response: Thanks for your comment. In this study, we confirm that the Sr/Ir ratio of the SI sample is slightly higher than 1 after acid washing. To dispel your doubts, we further performed following experiments. Considering the acid resistance of SI-

based series catalysts, we employed the aqua regia and reverse aqua regia dissolution methods. Furthermore, we have extended the dissolution time by threefold and conducted multiple ICP-MS experiments. ICP-MS test results of the sample SI dissolved in aqua regia and reverse aqua regia are presented in **Figure R3**. It can be observed that the Sr/Ir ratio in sample SI decreased from 1.17 (dissolved in aqua regia for 1 week) to 1.12 (dissolved in aqua regia for 3 weeks) and 1.06 (dissolved in reverse aqua regia for 3 weeks). So, after sufficient acid washing, the Sr/Ir ratio of SI sample was still slightly higher than 1 in this study.

Fig. R3 (a) ICP-MS diagram of Co/Ir and Sr/Ir ratios of sample SI. (b) Schematic diagram of sample SI structure.

Fig. R4 HAADF image and O K-edge EELS spectra of SI sample.

Considering that the Sr/Ir ratio of sample SI exceed 1.0, we conducted additional discussions. Based on the EDS mapping of the sample SI (Figure S4 in the supporting information), we found that Sr deficiency exists in the surface layer of the material at approximately 10 nm. But the bulk (about 500 nm) Sr proportion is higher than Ir as indicated by the EELS results (**Figure**

R4). The ICP-MS results are significantly influenced by the bulk Sr:Ir ratio, so “The Sr/Ir ratio of SI sample was slightly higher than 1” is understandable. We have also found the similar results in other paper, where the bulk structure of SrIrO₃ exhibits a Sr : Ir ratio higher than 1 (*Chem. Mater.*, 2020, 32, 4509). It is worth mentioning that many studies only rely on the results of XPS or EDS characterizations to determine the elemental ratio (*Nat. Commun.*, 2018, 9, 5236; *Appl. Catal. B Environ.*, 2021, 298, 120562; *ACS Appl. Energy Mater.*, 2022, 5, 6146), but these characterizations are significantly influenced by the surface elements composition of the catalysts. Relatively speaking, the ICP-MS is an accurate method for determining the bulk elements composition. We have provided this explanation in the manuscript. The Sr/Ir ratio of the sample SI was slightly higher than 1, indicating a slightly higher Sr proportion compared to Ir in bulk structure, as shown in the EELS results in **Figure R4**.

Fig. R5 (a) ICP-MS diagram of Co/Ir and Sr/Ir ratios of SI₁C₁. (b) ICP-MS diagram of Co/Ir and Sr/Ir ratios of SI₂C₁.

Fig. R6 (a) ICP-MS diagram of Co/Ir and Sr/Ir ratios of SI₄C₁ sample. (b) ICP-MS diagram of Co/Ir and Sr/Ir ratios of SI₆C₁.

According to your comment, we also dissolved other samples, such as SI₁C₁, SI₂C₁, SI₄C₁, SI₆C₁ and SI₈C₁, by using aqua regia/reverse aqua regia and conducted ICP-MS testing, and the results are shown in **Figures R5-R7**, which show the Sr/Ir ratio

significantly increased with the decrease of Co doping. Among above samples, the sample SI_1C_1 dissolved by reverse aqua regia 3 weeks displayed the lowest Sr/Ir ratio of 0.22 as shown in **Figure R5**, which is much lower than 1. From **Figure R7b**, it can be observed that the Sr/Ir ratios of SI_1C_1 , SI_2C_1 , SI_4C_1 , SI_6C_1 and SI_8C_1 all are lower than that of sample SI (>1). This result suggests that the doping of Co slightly reduces the strong oxidizing acid tolerance of sample SI. The above change trend of Sr/Ir ratio in different samples is consistent with our original paper, and we have provided the updated results in the revised paper.

Fig. R7 (a) ICP-MS diagram of Co/Ir and Sr/Ir ratios of SI_8C_1 . (b) ICP-MS diagram of Co/Ir and Sr/Ir ratios of various samples.

Comment 2: In general, the activity is enhanced will lead to a decreasing stability, did the authors test the stability of catalysts in solution of three-electrode system? It is stable or unstable compare to commercial IrO_2 ? I guess that Co-doped $SrIrO_3$ will not too stable in in solution of three-electrode system. Why it will extend a few hundred hours in PEM test?

Response: Thanks for your comment. We carefully answered your comments and conducted the additional experiments.

For your comment of “*In general, the activity is enhanced will lead to a decreasing stability, did the authors test the stability of catalysts in solution of three-electrode system? It is stable or unstable compare to commercial IrO_2 ?*”, our response is as follows: The improvement in performance does not necessarily indicate a decrease in stability, as extensively reported in previous references (*Science*, 2016, 353, 1011; *Nat. Commun.*, 2022, 13, 7935; *Nat. Commun.*, 2019, 10, 572; *Angew. Chem. Int. Ed.*, 2023, 62, e202311606.). We have conducted the stability tests by using a three-electrode system, and the results are shown in **Figure R8**. It can be observed that the sample SI_6C_1 with higher activity exhibits higher stability than the SI sample at 10 mA/cm², with overpotential increasing of only 28.2 mV over 100 h (for sample SI, the overpotential increases 32.0 mV over 100 h). Additionally, the stability of the sample SI_6C_1 is comparable to that of commercial IrO_2 .

For your comment of “*I guess that Co-doped $SrIrO_3$ will not too stable in solution of three-electrode system*”, our response is

as follows: The catalytic performance and stability of catalysts may not necessarily be directly related. Many reports on metal-doped perovskite catalysts show that the dissolution of metal elements does not necessarily lead to a decrease in performance and stability (*Nat. Commun.*, 2019, 10, 572; *Angew. Chem. Int. Ed.*, 2023, 62, e202311606; *Nat. Catal.*, 2019, 2, 763-772; *Angew. Chem.*, 2019, 131, 4619). In this study, low levels of Co doping do not significantly impact the stability of the catalyst because catalysis primarily occurs on the surface of catalyst, which is mainly composed of unsaturated IrO_x . Even after an extensive OER catalysis process, the unsaturated IrO_x may eventually be filled with O, and the resulting saturated IrO_x still exhibits high performance comparable to the sample SI as shown in **Figure R8**. To demonstrate this viewpoint, we have also supplement new XAS data after OER, as shown in **Figures R9-R11**.

Fig. R8 Stability tests of various samples were conducted by using a three-electrode system (@10 mA/cm²).

Fig. R9 (a) $\chi(R)$ space spectra fitting curve of SI_6C_1 (after 24 h of CV). (b) $k^3\chi(k)$ space spectra fitting curve of SI_6C_1 (after 24 h of CV). (c) $\chi(R)$ space spectra fitting curve of SI (after 24 h of CV). (d) $k^3\chi(k)$ space spectra fitting curve of SI (after 24 h of CV).

Fig. R10 (a) Changes in Ir-O coordination of SI sample (after 24 h of CV). (b) Changes in Ir-O coordination of SI₆C₁ sample (after 24 h of CV).

Based on the above results of XAS data, it can be observed that the Ir-O coordination number of the sample SI₆C₁ slightly increased after 24 hours of CV, indicating a significant LOPAEM process and suggesting high structural stability for unsaturated IrO_x. On the other hand, the Ir-O coordination number of the sample SI was increased, which can be attributed to the filling of oxygen during the reaction process (*Sci. Adv.*, 2021, 7, eabc7323), leading to an increase in the Ir-O coordination number of the catalyst. In addition, we also conducted XAS testing on sample SI₆C₁ after 100 h of CV as shown in **Figure R11**. We observed a slight increase in the coordination number, which can be attributed to high rate of oxygen filling during the reaction process compared to the rate of LOM.

Fig. R11 (a) $\chi(R)$ space spectra fitting curve of SI₆C₁ (after 100 h of CV). (b) $k^3\chi(k)$ space spectra fitting curve of SI₆C₁ (after 100 h of CV).

Table R1. EXAFS fitting parameters of samples at the Ir L-edge.

Sample	path	C.N.	R(Å)	$\sigma^2(\times 10^{-3}\text{Å}^2)$	$\Delta E(\text{eV})$	R factor
SI ₆ C ₁ 100 h	Ir-O	5.627	2.01	4.6	9.0	0.86%
SI ₆ C ₁ 24 h	Ir-O	5.446	2.01	5.4	9.0	0.66%
SI 24 h	Ir-O	5.743	1.99	6.6	7.8	0.94%

Fig. R12. PEM water electrolysis stability of SI₆C₁ (at 1000 mA/cm²).

For your comment of “*Why it will extend a few hundred hours in PEM test?*”, our response is as follows: We have conducted numerous PEMWE experiments and found that the catalytic stability of the sample SI₆C₁ is excellent even at a high current density of 1000 mA/cm² for 1000000s as shown in **Figure R12**. The sample SI₆C₁ can extend a few hundred hours of PEM because the unsaturated IrO_x structure is relative stable during OER. SI₆C₁ is mainly undergoes AEM, and we have proved that the unsaturated IrO_x after long term OER catalysis is still stable by XAS as shown in **Figure R11**. In addition, many reports on metal-doped systems show that the dissolution of metal elements does not necessarily lead to a decrease in performance and stability (*Nat. Commun.*, 2019, 10, 572; *Angew. Chem. Int. Ed.*, 2023, 62, e202311606; *Nat. Catal.*, 2019, 2, 763-772; *Angew. Chem.*, 2019, 131, 4619). Furthermore, in a recent study (published 2 months ago, *Angew. Chem. Int. Ed.*, 2023, e202313954), commercial IrO₂ was treated with plasma, leading to the formation of low-coordinated IrO_x catalyst, and the low-coordinated IrO_x catalyst demonstrates high activity and exceptional stability, even surpassing 1000 hours for PEMWE, which is consistent with our results.

Comment 3: There are errors in the annotations of Figures 1, S10, S11, S14, S15, S16, and S17. In Figure 1, the d is missing.

In Figure S10 and 11, d, e, f is missing. In S14, S15, S16 c, d is missing. In Figure S17, the c, d, e, f is missing.

Response: Thanks for your careful comment. We have revised the above errors according to your comments, and the corrections are listed as follows:

Figure 1. Characterizations of samples. (a) OER catalytic mechanism diagram of Co-doped SrIrO₃ catalyst. (b-d) HRTEM diagram and corresponding FFT diagram of SI, SI₆C₁ and SI₁C₁. (e) O K-edge EELS spectra of SI₆C₁ (f) HAADF image of SI₆C₁ and corresponding EDS mapping images. (g-h) XRD spectrum of SI series samples. (i) ICP-MS diagram of Co/Ir and Sr/Ir ratios of SI series samples.

Figure S10. The ¹⁶O¹⁶O and ¹⁸O¹⁶O intensities of (a, d) SI₁C₁ (b, e) SI₂C₁ and (c, f) SI₄C₁ tested by DEMS.

Figure S11. The ¹⁶O¹⁶O and ¹⁸O¹⁶O intensities of (a, d) SI₆C₁ (b, e) SI₈C₁ and (c, f) SI tested by DEMS.

Figure S14. (a, c) $\chi(R)$ space spectra fitting curves of SI (ex-situ and ocp). (b, d) $k^3\chi(k)$ space spectra fitting curves of SI (ex-situ and ocp).

Figure S15. (a, c, e) $\chi(R)$ space spectra fitting curves of SI (1.033 V, 1.233 V and 1.333 V vs. RHE). (b, d, f) $k^3\chi(k)$ space spectra fitting curves of SI (1.033 V, 1.233 V and 1.333 V vs. RHE).

Figure S16. (a, c) $\chi(R)$ space spectra fitting curves of SI₆C₁ (ex-situ and ocp). (b, d) $k^3\chi(k)$ space spectra fitting curves of SI₆C₁ (ex-situ and ocp).

Figure S17. (a, c, e) $\chi(R)$ space spectra fitting curves of SI₆C₁ (1.033 V, 1.233 V and 1.333 V vs. RHE). (b, d, f) $k^3\chi(k)$ space spectra fitting curves of SI₆C₁ (1.033 V, 1.233 V and 1.333 V vs. RHE).

Comment 4: The authors should compare the stability of SI for PEM with the prepared catalysts.

Response: Thanks for your excellent comment. We have compared the stability of SI for PEM with the prepared catalysts, such as SI₈C₁ and SI₁C₁, at 1000 mA/cm² as shown in **Figures R13-15**.

Fig. R13 PEM water electrolysis stability of SI (at 1000 mA/cm²).

Fig. R14 PEM water electrolysis stability of SI_8C_1 (at 1000 mA/cm^2).

Fig. R15 PEM water electrolysis stability of SI_1C_1 (at 1000 mA/cm^2).

Reply to the Comments of Reviewer #2:

General comments: This manuscript presents a study on electrocatalytic water splitting, focusing on a Co-doped SrIrO_3 system for enhancing the oxygen evolution reaction (OER) in an acidic medium. The study looks into how SrIrO_3 -based perovskite-type catalysts get their high catalytic activity. This is an important part of making hydrogen in a way that does not harm the environment. The authors have employed a combination of in-situ experiments and theoretical calculations to elucidate the role of Co doping in activating surface oxygen and optimizing adsorbate binding energy. The findings significantly contribute to our understanding of the catalytic mechanisms in acidic OER and offer valuable insights for designing efficient electrocatalysts. However, the study also presents certain limitations and areas that require clarification to bolster the presented conclusions.

Response: We sincerely appreciate your positive comments on our paper. Your excellent professional opinions and evaluations have a constructive impact on improving this manuscript. Your positive comments also strengthened our confidence in scientific research. Based on your comments, we have made substantial revisions to the manuscript by adding experiments and discussions to fully address your concerns. Thank you very much.

Comment 1: The authors should fix the problem with the Sr:Ir ratio in SI_6C_1 and SI samples that was pointed out in Line 84. There is a difference between the EELS results and the compound ratios. Clarification on this matter is crucial, as it directly impacts the understanding of the catalyst composition and its subsequent catalytic behavior.

Response: Thank you for your attentive and helpful comment. According to your comment, we clarify the difference between EELS results and compound ratios and explain it as follows:

To investigate the reason for the diminished crystallinity observed in SI_6C_1 sample, we have conducted electron energy loss spectroscopy (EELS) analyses, and the results are presented in **Figure R16**. The numbers 1-11 in HAADF map correspond to the different positions, and 1-11 EELS spectra are measured correspond to the positions 1-11 in HAADF map, respectively. The numbers 1-11 in EELS spectra are the serial numbers of EELS spectra, and they do not represent different Co:Ir ratios. Due to the different measurement positions in SI_6C_1 sample, the measured EELS spectra are different. We compared the EELS of the SI_6C_1 sample with the supporting SI sample. The EELS result does not reflect the overall element ratio, but rather the gradient element ratio (the difference between surface and bulk proportions). The compound ratios in this study were obtained by ICP-MS measurement, and the results of ICP-MS measurements represent the element ratios in the overall samples. So, there is not a strong correlation between the EELS results and the compound ratios in SI_6C_1 and SI samples. We also noticed that the signal of SI sample was weak, and we supplemented the EELS data of SI as shown in **Figure R17**. It can be observed that the Sr peak of SI is higher than that of SI_6C_1 , which is consistent with the conclusion in our paper.

Fig. R16 HAADF image and O K-edge EELS spectra of SI sample.

Fig. R17 (a) surface O K-edge EELS spectra of SI sample and SI_6C_1 sample; (b) bulk O K-edge EELS spectra of SI sample and SI_6C_1 sample.

Comment 2: The absence of peaks in the XRD pattern of sample SI_1C_1 , as shown in Figure 1g, raises questions about the phase of the material. The authors should address whether the consistent doping ratio of Ir and Co indeed leads to an amorphous phase, and if so, discuss the implications of this phase on the catalyst's performance.

Response: Thank you very much for the professional comment. We have had a thorough discussion on this issue. Firstly, the decreasing crystallinity of the catalyst with Co doping is attributed to the difficulty in maintaining the SrIrO_3 phase, especially for sample SI_1C_1 . However, when the SI_1C_1 was subjected to heat treatment at a temperature of 900°C ($\text{SI}_1\text{C}_1\text{-}900$), we observed a slight improvement in the crystallinity of the sample $\text{SI}_1\text{C}_1\text{-}900$ as shown in **Figure R18a**, which clearly shows the existence of the phase of SI_1C_1 for sample $\text{SI}_1\text{C}_1\text{-}900$. So, we are confident that the consistent doping ratio of Ir and Co does indeed lead to amorphous phase of SI_1C_1 at room temperature.

To discuss the implications of this phase on the catalyst's performance, we compared the OER catalytic performance of SI_1C_1 with different crystallinities as shown in **Figure R18b**, which shows a small difference between samples SI_1C_1 and $\text{SI}_1\text{C}_1\text{-}900$. This suggests that the crystallinity of SI_1C_1 has a small effect on OER catalytic performance. To clarify this, we further discussed the origins of the OER catalytic activity of SI_1C_1 sample. Firstly, we analyzed HRTEM image of SI_1C_1 sample and corresponding FFT diagrams of the different sites as shown in **Figure R19**. It can be observed that the sites of 3, 4, 5 on the surface of SI_1C_1 exhibits a short-range ordered structure. In addition, HRTEM images of SI_6C_1 sample after OER are shown in **Figure R20b**, which shows that a small amount of amorphous phase appeared on the surface of SI_6C_1 sample after OER. The formation of the unsaturated IrO_x may contribute to the presence of the amorphous layer. The SI_1C_1 and other Co-doped SrIrO_3 samples exhibit similar OER catalytic activities as shown in Figure 2 in the paper and own similar OER catalytic mechanisms that were proposed

as illustrated in **Figure R21**. In this study, whether it is Si_1C_1 or Si_6C_1 , initially, the IrO_6 structure undergoes surface reconstruction under LOPAEM for OER, ultimately leading to the formation of active species with unsaturated IrO_x structures. Therefore, the implications of the phase of Si_1C_1 on the catalyst's OER performance is not obvious.

Fig. R18 (a) XRD patterns of Si_1C_1 samples. (b) OER polarization curves of Si_1C_1 and Si_1C_1 -900.

Fig. R19 HRTEM diagram and corresponding FFT diagrams of Si_1C_1 .

Fig. R20 (a) HRTEM diagram and corresponding FFT diagram of SI_6C_1 . (b) HRTEM diagram and corresponding FFT diagram of SI_6C_1 after 24 h of CV for OER.

Fig. R21 Proposed possible OER reconstruction mechanism of SI_1C_1 and SI_6C_1 .

Comment 3: In the computational section, it is essential for the authors to provide details on the catalyst structure built for the SI_6C_1 system. Information about the dissolved proportions of Sr, Co, and O in the model is needed. Additionally, the authors

should explain how surface reconstruction, promoted by Co dissolution as mentioned in lines 175 and 189, is incorporated in the simulation model.

Response: Thank you very much for the professional comment. We have sorted your comments into two questions and replied them one by one.

For your comment of “*In the computational section, it is essential for the authors to provide details on the catalyst structure built for the SI₆C₁ system. Information about the dissolved proportions of Sr, Co, and O in the model is needed.*”, our response is as follows: Thank you very much for your comment on catalyst structure. I have provided detailed information about the DFT computational models for SI and SI₆C₁, as shown in **Tables R2** and **R3**.

Table R2. Structure parameters of SI (IrO_x) sample.

number	atom	x	y	z
1	O	0.20557	0.70398	0.5042
2	O	0.90536	0.0801	0.49123
3	O	0.64418	0.29791	0.46876
4	Ir	0.48304	0.32046	0.51018
5	Ir	0.85463	0.23456	0.44221
6	Ir	0.41347	0.71459	0.48651
7	Ir	0.8116	0.9152	0.45636
8	O	0.34732	0.20218	0.54002
9	O	0.50108	0.54555	0.52736
10	O	0.64032	0.77596	0.47423
11	O	0.89174	0.78112	0.40997
12	O	0.35165	0.89026	0.44029
13	O	0.09315	0.23647	0.42666
14	O	0.17831	0.04684	0.35833
15	O	0.85373	0.4044	0.3929
16	O	0.57834	0.58418	0.39449
17	Ir	0.39885	0.06479	0.33371
18	Ir	0.01983	0.23079	0.36755
19	Ir	0.38292	0.71554	0.40035
20	Ir	0.80095	0.62456	0.36824
21	O	0.19768	0.61302	0.38559
22	O	0.46227	0.84337	0.35087
23	O	0.54379	0.18906	0.36181
24	O	0.9246	0.23583	0.31219
25	O	0.3176	0.27604	0.30046
26	O	0.77342	0.75994	0.31842

27	O	0.87064	0.57366	0.25079
28	O	0.69541	0.02741	0.27111
29	O	0.35321	0.98854	0.27584
30	Ir	0.23377	0.18142	0.24899
31	Ir	0.84336	0.20728	0.25336
32	Ir	0.10769	0.60207	0.2428
33	Ir	0.7722	0.80081	0.25888
34	O	0.03698	0.05358	0.24642
35	O	0.06533	0.36273	0.2361
36	O	0.21345	0.62917	0.29227
37	O	0.77056	0.80528	0.19965
38	O	0.20047	0.74254	0.20226
39	O	0.34227	0.19133	0.19644
40	O	0.73247	0.22679	0.20067
41	O	0.00267	0.28356	0.15087
42	O	0.2972	0.50028	0.14793
43	O	0.70814	0.50028	0.14793
44	O	0.50267	0.79917	0.14676
45	Ir	0.25267	0.25000	0.13793
46	Ir	0.75267	0.25000	0.13793
47	Ir	0.25267	0.75000	0.13793
48	Ir	0.75267	0.75000	0.13793
49	O	0.00267	0.70083	0.12911
50	O	0.7972	0.99972	0.12794
51	O	0.20814	0.99972	0.12794
52	O	0.50267	0.21644	0.125
53	Sr	0.50267	0.48422	0.07899
54	Sr	0.50267	0.97412	0.07633
55	O	0.7972	0.28801	0.07226
56	O	0.20814	0.28801	0.07226
57	O	0.2972	0.71199	0.07211
58	O	0.70814	0.71199	0.07211
59	Sr	0.00267	0.02588	0.06803
60	Sr	0.00267	0.51578	0.06538

Table R3. Structure parameters of Sr_6C_1 ($\text{Ir}_{\text{unsat}}\text{Co}_2\text{L}\text{O}_x$) sample.

number	atom	x	y	z
1	O	0.29249	0.50063	0.42038
2	O	0.82446	0.13579	0.49826
3	O	0.64228	0.40184	0.46599
4	Ir	0.58029	0.46498	0.52287

5	Ir	0.82491	0.26266	0.44233
6	Ir	0.68981	0.99281	0.465
7	O	0.48306	0.31202	0.55216
8	O	0.63196	0.66914	0.53321
9	O	0.47475	0.99999	0.47171
10	O	0.78868	0.82211	0.44098
11	O	0.056	0.18636	0.4296
12	O	0.13601	0.01615	0.35745
13	O	0.87571	0.4141	0.39434
14	O	0.60842	0.54823	0.36781
15	Ir	0.35587	0.03583	0.33466
16	Ir	0.99012	0.20563	0.37016
17	Ir	0.39927	0.6153	0.38242
18	Co	0.8273	0.60348	0.36032
19	O	0.01499	0.68003	0.36539
20	O	0.39433	0.81194	0.35481
21	O	0.51989	0.13211	0.36424
22	O	0.90468	0.22378	0.31377
23	O	0.30174	0.25608	0.30563
24	O	0.76932	0.75191	0.3161
25	O	0.86984	0.5713	0.24864
26	O	0.68285	0.02242	0.26792
27	O	0.349	0.97345	0.27504
28	Ir	0.23024	0.17647	0.25013
29	Ir	0.83645	0.20259	0.25426
30	Ir	0.1066	0.59889	0.24197
31	Ir	0.76621	0.79791	0.25865
32	O	0.03217	0.05259	0.24644
33	O	0.06491	0.35961	0.23887
34	O	0.21481	0.62818	0.29091
35	O	0.76719	0.80552	0.19968
36	O	0.19786	0.74206	0.20187
37	O	0.33676	0.18955	0.19691
38	O	0.73352	0.22433	0.20039
39	O	0.00267	0.28356	0.15087
40	O	0.2972	0.50028	0.14793
41	O	0.70814	0.50028	0.14793
42	O	0.50267	0.79917	0.14676
43	Ir	0.25267	0.25000	0.13793
44	Ir	0.75267	0.25000	0.13793
45	Ir	0.25267	0.75000	0.13793
46	Ir	0.75267	0.75000	0.13793
47	O	0.00267	0.70083	0.12911

48	O	0.7972	0.99972	0.12794
49	O	0.20814	0.99972	0.12794
50	O	0.50267	0.21644	0.125
51	Sr	0.50267	0.48422	0.07899
52	Sr	0.50267	0.97412	0.07633
53	O	0.7972	0.28801	0.07226
54	O	0.20814	0.28801	0.07226
55	O	0.2972	0.71199	0.07211
56	O	0.70814	0.71199	0.07211
57	Sr	0.00267	0.02588	0.06803

For your comment of “*Additionally, the authors should explain how surface reconstruction, promoted by Co dissolution as mentioned in lines 175 and 189, is incorporated in the simulation model.*”, our response is as follows: Thank you very much for your comment on surface reconstruction. Through DFT calculations, we discovered that the direct doping of Co could not enhance the OER performance of the catalyst. During the research process, we demonstrated through experiments that the improvement in OER catalytic performance was attributed to the formation of unsaturated IrO_x structures after Co dissolution. Subsequently, we performed DOS calculations and found that Co could modulate the O 2p-band center of IrO_x on the surface, destabilizing the surface oxygen of the catalyst, thereby facilitating lattice oxygen oxidation (*Nat. Chem.*, 2017, 9, 457-65; *Nat. Commun.*, 2016, 7, 11053; *Nat. Commun.*, 2013, 4, 2439; *Nat. Catal.*, 2019, 2, 763-72).

Comment 4: Regarding Line 297, a clearer explanation of the main differences and advantages of the newly proposed OER mechanism (LOPAEM) compared to traditional mechanisms (AEM and LOM) is necessary. This clarification will help in understanding the novelty and significance of the proposed mechanism in the broader context of OER research.

Response: Thank you very much for the professional comment. As the mechanism section was difficult to comprehend through conventional images and text, we created an academic animation to visually illustrate the mechanism of our work. This will provide a clearer explanation of the main differences and advantages of the LOPAEM mechanism compared to the traditional mechanisms (AEM and LOM). The animation is included as **Supplementary Movie**. Additionally, we have provided additional supplementary information on LOPAEM mechanism as shown below:

Unlike the conventional mechanisms of OER, LOPAEM mechanism involves the synergistic action of both mechanisms. Specifically, when the oxidation rate of lattice oxygen in certain catalysts becomes excessively fast, it leads to the formation of more surface oxygen vacancies (i.e., unsaturated metal sites), thereby altering the original catalytic sites of the catalyst. These unsaturated metal sites may exhibit more efficient AEM performance, thus achieving LOPAEM mechanism.

Reply to the Comments of Reviewer #3:

General comments: This study reveals the origin of Co doping in SrIrO₃ for acidic OER, which results in surface unsaturated IrO_x. Although some literatures have already reported the application and the dissolution of Co-doped SrIrO₃, this study provides a deeper analysis and understanding. Objectively speaking, its OER activity, such as the overpotential at 10 mA cm⁻², is moderate, especially compared to other recently-reported catalysts. There are several issues still needed to be addressed before publication. I suggest that the current manuscript requires a major revision.

Response: We sincerely appreciate your positive comments on our paper. Your comments is highly valuable for us. Accordingly, we have made substantial revisions in our manuscript to fully address your concerns by adding experimental with discussions. We appreciate your comments on the OER activity. Indeed, most of the recently reported highly active Ir-based catalysts often involve specific nanostructural designs such as layered nanostructures, core-shell nanoarchitectures, and so on (*Nat. Commun.*, 2023, 14, 5119; *Nat. Commun.*, 2023, 14, 1248; *Nat. Commun.* 2021, 12, 3540; *Energy Environ. Sci.*, 2023, 16, 3734). In our study, we primarily focused on investigating the OER catalytic mechanism of Co-doped SrIrO₃ catalyst and did not intentionally control the structure or morphology. We acknowledge that the performance is not particularly outstanding, and in future research, we will make great efforts to further improve and optimize OER catalytic performance. Thank you very much for providing us with valuable insights and the opportunity to revise our paper.

Comment 1: The introduction of Co results in surface unsaturated IrO_x. Whether the surface unsaturated IrO_x remains stable during acidic OER operation. The material degradation is also crucial for the commercial application. Please provide the TEM image after long-term operation.

Response: Thank you very much for the professional comment. Yes, the introduction of Co results in surface unsaturated IrO_x. In this study, the surface unsaturated IrO_x remains high stable during the acidic OER operation. According to your comments, we provided the TEM image of catalyst after long-term OER catalysis. HRTEM images of SI₆C₁ and SI catalysts after OER are shown in **Figure R22** and **R23**, respectively. We found that a small amount of amorphous phase appeared on the surface of the SI₆C₁ sample after prolonged OER as shown in Figure R23. The formation of unsaturated IrO_x contributes to the presence of the amorphous layer (*Angew. Chem. Int. Ed.*, 2023, e202313954). We also found that amorphous phase appeared on the surface of the SI sample after OER as shown in Figure R24, which can be attributed to the formation of saturated IrO_x structures. To further determine the stability of unsaturated IrO_x, the SI₆C₁ and SI catalysts after OER were also characterized by using XAS, which can provide a direct response to your comment as shown in **Figures R24-25** and **Table R4**.

From the results shown in **Figures R24-25**, it can be observed that the Ir-O coordination number of the SI_6C_1 sample remains nearly unchanged after 24 hours of CV (Table R4), indicating a significant LOPAEM process and suggesting good structural stability for the unsaturated IrO_x . On the other hand, the Ir-O coordination number of the SI sample increased after 24 hours of CV, which can be attributed to the filling of oxygen during the reaction process, leading to an increase in the Ir-O coordination number of the catalyst (*Sci. Adv.*, 2021, 7, eabc7323). In addition, we also conducted XAS testing on the SI_6C_1 sample after 100 h of CV as shown in **Figure R26**. We observed a slight increase in the coordination number (Table R4), which could be attributed to higher rate of oxygen filling during the reaction process compared to the rate of LOM.

Fig. R22 (a) HRTEM image and corresponding FFT diagram of SI_6C_1 ; (b) HRTEM image and corresponding FFT diagram of SI_6C_1 after 24 h CV for OER.

Fig. R23 (a) HRTEM image and corresponding FFT diagram of SI; (b) HRTEM image and corresponding FFT diagram of SI after 24 h CVs for OER.

Fig. R24 (a) $\chi(R)$ space spectrum fitting curve of SI_6C_1 (after 24 h of CV); (b) $k^3\chi(k)$ space spectrum fitting curve of SI_6C_1 (after 24 h of CV); (c) $\chi(R)$ space spectrum fitting curve of SI (after 24 h of CV); (d) $k^3\chi(k)$ space spectrum fitting curve of SI (after 24 h of CV)

Fig. R25 (a) Changes in Ir-O coordination of SI sample (after 24 h of CV); (b) Changes in Ir-O coordination of SI_6C_1 sample (after 24 h of CV).

Fig. R26 (a) $\chi(R)$ space spectrum fitting curve of SI_6C_1 (after 100 h of CV). (b) $k^3\chi(k)$ space spectrum fitting curve of SI_6C_1 (after 100 h of CV).

Table R4. EXAFS fitting parameters of samples at the Ir L-edge.

Sample	path	C.N.	R(Å)	$\sigma^2(\times 10^{-3}\text{Å}^2)$	$\Delta E(\text{eV})$	R factor
SI_6C_1 100 h	Ir-O	5.627	2.01	4.6	9.0	0.86%
SI_6C_1 24 h	Ir-O	5.446	2.01	5.4	9.0	0.66%
SI 24 h	Ir-O	5.743	1.99	6.6	7.8	0.94%

Comment 2: During DFT calculation, how are the representative models constructed. There are many possibilities for the surface unsaturated IrO_x with Co and Sr leaching.

Response: Thank you very much for the professional comment. During the research process, we have demonstrated through experiments that the improvement in OER performance was attributed to the formation of unsaturated IrO_x structures after Co dissolution. We provided the construction of unsaturated IrO_x models for DFT calculations as shown in **Figure R27**.

Firstly, we constructed and optimized the slab structure of the original SrIrO_3 . Next, we remove the surface Sr of the catalyst. According to reference and characterization results (*Sci. Adv.*, 2021, 7, eabc7323), we found that the Ir-O coordination number on the catalyst surface is approximately 4.5, so removing 4 Sr atoms will remove 3 neighboring O atoms. The same method applies to the Co-doped system, except that removing Co will take away 2 neighboring O atoms. We acknowledge that there are many possible ways to construct the system, but it is not feasible to calculate each one. To address your comment, we also refer to the method proposed by Nørskov *et al.* (*Science*, 2016, 353, 1011) and directly remove Sr. The results of the free energy calculations are shown in **Figure R28**.

From the results of the free energy calculations, it can be observed that for most models, the rate-determining step (RDS) remains as the OOH step. However, there has been a significant change in the adsorption mode of $\text{IrCo}_{21}\text{O}_x$, where its RDS has

shifted to OH. Additionally, the $\text{Ir}_{\text{unsat}}\text{Co}_2\text{L}\text{O}_x$ model also exhibits the lowest overpotential, consistent with the calculations presented in the manuscript.

In addition, we also calculated the density of states (DOS) for different models, as shown in **Figure R29**. We found that Co doping and the formation of unsaturated IrO_x favored the positively shift of O-2p band center, thereby facilitating lattice oxygen oxidation (*Nat. Chem.*, 2017, 9, 457-65; *Nat. Commun.*, 2016, 7, 11053; *Nat. Commun.*, 2013, 4, 2439; *Nat. Catal.*, 2019, 2, 763-72), which is consistent with the conclusions drawn in our manuscript. The lower Ir-3d band center can also be observed from the results, highlighting the importance of Ir-3d states for efficient AEM.

Fig. R27 Schematic diagram of the construction method for IrO_x structure.

Fig. R28 OER free energy diagrams of different models (without leaching of O): (a) IrO_x structure. (b) IrCo_2LO_x . (c) $\text{Ir}_{\text{unsat}}\text{O}_x$. (d) $\text{Ir}_{\text{unsat}}\text{Co}_2\text{LO}_x$.

Fig. R29 Density of states diagrams for (a) IrO_x , (b) $\text{IrCo}_{2\text{L}}\text{O}_x$, (c) $\text{Ir}_{\text{unsat}}\text{O}_x$, (d) $\text{Ir}_{\text{unsat}}\text{Co}_{2\text{L}}\text{O}_x$.

Comment 3: In the case of in-situ Raman, are there any other key Ir-related reaction intermediates observed?

Response: Thank you very much for the professional comment. We carefully examined the results of in-situ Raman as shown in **Figures R30-31**. We found that the SI and SI_6C_1 samples did not show any significant presence of other Ir-related intermediate species. We think that the reason for this observation could be attributed to the SrIrO_3 structure (*ACS Catal.*, 2016, 6, 8098; *J. Raman Spec.*, 2007, 38, 737), where even during the OER catalytic process, it predominantly exists in the form of Ir- μ -ox stretching vibration. To confirm this hypothesis, we conducted additional in-situ Raman experiments for the SI_1C_1 sample. The results revealed that no other Ir intermediates were observed.

Fig. R30 In-situ Raman spectra of SI sample and SI_6C_1 sample.

Fig. R31 In-situ Raman spectra of SI_1C_1 sample.

Comment 4: In fig3d, the colors of the curves are difficult to distinguish.

Response: Thank you very much for the professional comment. According to your comment, we changed the colors of the curves in Figure 3d. However, the change of the colors of the curves in Figure 3d still doesn't allow for clear differentiation of the curves because the different Co doping shows a minimal impact on the free energy as shown in **Figure R32**. To address this

issue, we created separate free energy diagrams for each model as shown in **Figure R33**, and this figure is also shown in Figure S27 in the supporting information.

Fig. R32 OER free energy diagrams of different models.

Fig. R33 OER free energy diagrams of different models. (a) IrO_x , (b) $\text{IrCo}_{4\text{L}}\text{O}_x$, (c) $\text{IrCo}_{3\text{L}}\text{O}_x$, (d) $\text{IrCo}_{2\text{L}}\text{O}_x$, (e) $\text{IrCo}_{\text{surf}}\text{O}_x$, (f) $\text{Ir}_{\text{unsat}}\text{Co}_{2\text{L}}\text{O}_x$ and (g) $\text{Ir}_{\text{unsat}}\text{Co}_{2\text{L}}\text{O}_x$.

REVIEWERS' COMMENTS

Reviewer #1 (Remarks to the Author):

The authors modified the manuscript well, it is suitable for publication now.

Reviewer #2 (Remarks to the Author):

The authors have fully addressed the concerns in the revised manuscript. I have no more comments.

Reviewer #3 (Remarks to the Author):

While the innovative aspect of your work is moderate, the depth of your research is commendable. Your detailed answers have successfully addressed my concerns. I recommend your manuscript for acceptance.